# Remodelling of the gut microbiota by hyperactive NLRP3 induces regulatory T cells to maintain homeostasis

Xiaomin Yao [1], Chenhong Zhang[2], Yue Xing[1], Guang Xue[1], Qianpeng Zhang[2], Fengwei Pan[2], Guojun Wu[2], Yingxin Hu[2], Qiuhong Guo[1], Ailing Lu[1], Xiaoming Zhang[1], Rongbin Zhou[3], Zhigang Tian[3], Benhua Zeng[4], Hong Wei[4], Warren Strober[5], Liping Zhao[2] & Guangxun Meng[1]

Inflammasomes are involved in gut homeostasis and inflammatory pathologies, but the role of NLRP3 inflammasome in these processes is not well understood. Cryopyrin-associated periodic syndrome (CAPS) patients with NLRP3 mutations have autoinflammation in skin, joints, and eyes, but not in the intestine. Here we show that the intestines of CAPS model mice carrying an $Nlrp3^{R258W}$ mutation maintain homeostasis in the gut. Additionally, such mice are strongly resistant to experimental colitis and colorectal cancer; this is mainly through a remodelled gut microbiota with enhanced anti-inflammatory capacity due to increased induction of regulatory T cells ($T_{regs}$). Mechanistically, $NLRP3^{R258W}$ functions exclusively in the lamina propria mononuclear phagocytes to directly enhance IL-1β but not IL-18 secretion. Increased IL-1β boosts local antimicrobial peptides to facilitate microbiota remodelling. Our data show that $NLRP3^{R258W}$-induced remodelling of the gut microbiota, induces local $T_{regs}$ to maintain homeostasis and compensate for otherwise-detrimental intestinal inflammation.

[1] Key Laboratory of Molecular Virology and Immunology, Institut Pasteur of Shanghai, Chinese Academy of Sciences; University of Chinese Academy of Sciences, Shanghai 200031, China. [2] State Key Laboratory of Microbial Metabolism, Department of Biological Sciences, School of Life Sciences and Biotechnology, Shanghai Jiao Tong University, Shanghai 200240, China. [3] Department of Immunology, School of Life Sciences, University of Science and Technology of China, Hefei, Anhui 230027, China. [4] Department of Laboratory Animal Science, College of Basic Medical Sciences, Third Military Medical University, Chongqing 400038, China. [5] Mucosal Immunity Section, Laboratory for Host Defenses, National Institute of Allergy and Infectious Diseases, National Institutes of Health, Bethesda, MD 20892, USA. Xiaomin Yao, Chenhong Zhang and Yue Xing contributed equally to this work. Correspondence and requests for materials should be addressed to L.Z. (email: lpzhao@sjtu.edu.cn) or to G.M. (email: gxmeng@ips.ac.cn)

The gastrointestinal tract is typically colonized with microbiota consisting of $10^{13}$–$10^{14}$ commensal microbes from hundreds of species, which need to interact with the host immune system to reach a "primed homeostasis" for maintaining gut health[1]. This homeostasis represents the most delicate balance in an organism in coping with endogenous/exogenous turbulences, in a manner that the host mounts a constant but flexible low-level inflammatory response while the microbiota exhibits a tuned structural plasticity[1, 2]. Failure to maintain the tonic inflammatory signaling can lead to dysbiosis, which may bring either spontaneous colitis or increased susceptibility to colitis[3–6]. As an important arm of the host innate immune system,

inflammasomes are involved in such inflammatory gut diseases[7]. Inflammasomes are protein complexes that consist of pattern recognition receptors, such as NLRP3 or AIM2, the adapter protein ASC, and the effector molecule pro-caspase-1[8]. The activation of inflammasome leads to auto-cleavage of caspase-1-mediating maturation of IL-1β and IL-18[8]. To avoid unnecessary inflammation-induced damage to host cells, inflammasomes are rigidly suppressed during the resting stage and are only activated upon appropriate triggering[9]. Genetic deficiencies in several inflammasome components have been shown to increase susceptibility to experimental colitis in mice[10–13], which is potentially due to the uncontrolled outgrowth of pathobionts that are

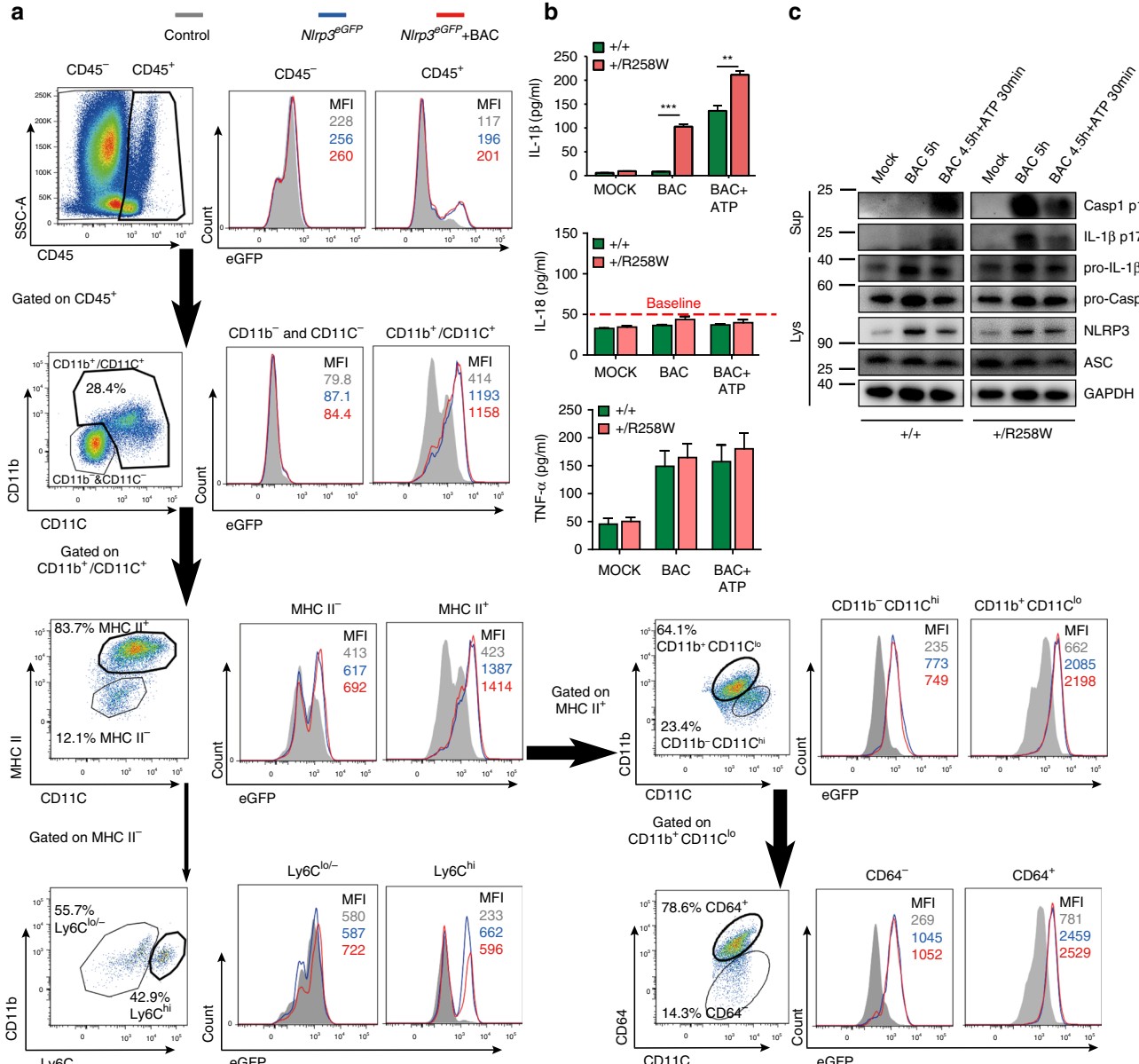

**Fig. 1** Intestinal lamina propria CD11b⁺/CD11C⁺ mononuclear phagocytes from *Nlrp3*^R258W mice contain hyperactive inflammasome. **a** Colonic lamina propria mononuclear cells (LPMCs) were isolated from control mice (gray filled) or *Nlrp3*^eGFP reporter mice left untreated (blue line), or stimulated with heat-inactivated fecal bacteria (BAC) at multiplicity of infection (MOI) = 20:1 for 4 h (red line); different cell populations were identified by the indicated surface markers and were examined for NLRP3-eGFP expression. MFI represents geomean fluorescence intensity of eGFP in each sample. **b, c** Colonic LPMCs were stimulated with BAC (MOI = 20:1) for 12 h (**b**) or 5 h (**c**) with or without a ATP pulse for 30 min; control cells (MOCK) were not stimulated. IL-1β, IL-18, and TNF-α in the supernatants were determined through ELISA. The data are representative of at least three independent experiments and are shown as the means ± SEM. **P < 0.01, ***P < 0.001 (Two-tailed Student's t test) (**b**); cleaved caspase-1 p10 subunit, IL-1β-p17 fragment in the supernatants (Sup), and their pro-forms together with NLRP3, ASC, and GAPDH in the cell lysates (Lys) were detected by western blot (**c**)

associated with chronic inflammatory conditions, including gut dysbiosis[10], [14]. However, a recent report showed that a gain-of-function mutation in NLRC4, known as V341A, causes hyper-active inflammasome activity and associated neonatal enterocolitis in patients[15]. This finding strongly suggests that a balanced inflammasome activity is critical for maintaining intestinal homeostasis. Interestingly, in humans, dysregulation of inflammasome signaling by gain-of-function mutations in the coding region of *NLRP3* results in cryopyrin-associated periodic syndromes (CAPS), which are characterized by auto-

inflammatory conditions in skin, joints, and eyes, while no gut symptoms were reported[16–20].

Notably, genetically modified mice carrying the *Nlrp3*[R258W] mutation, which is homologous to the human *NLRP3*[R260W], also develop similar auto-inflammatory conditions in the skin[21] in association with the presence of microbes[22, 23]. As the intestinal microbiota consists of trillions of microbes, enhanced inflammasome signaling by the *Nlrp3*[R258W] mutation is expected to pose a huge challenge for host maintenance of gut homeostasis. However, no overt inflammatory pathology is observed in the

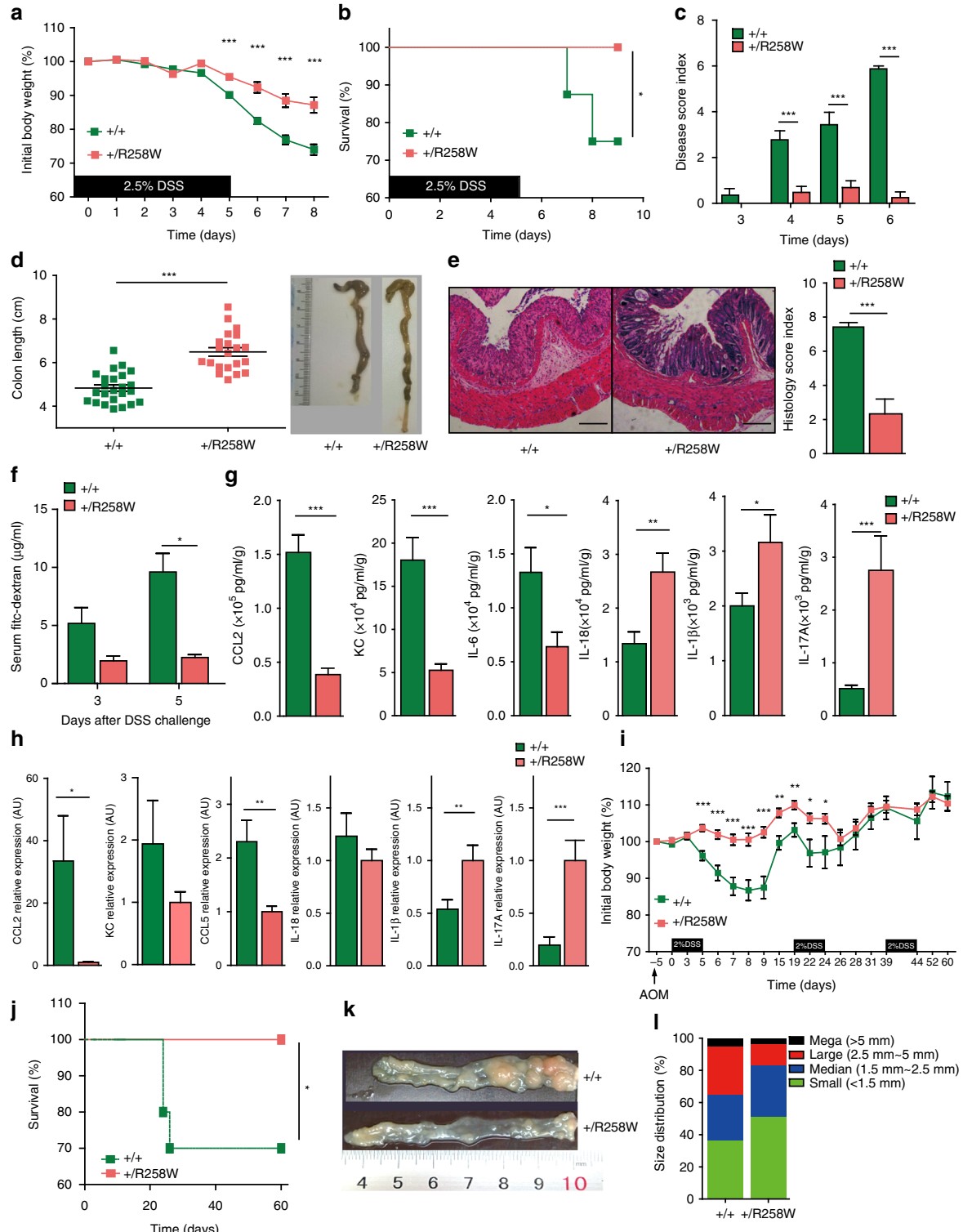

intestine of $Nlrp3^{R258W}$ mice[21]. This seemingly paradoxical phenomenon observed in the homeostatic guts of $Nlrp3^{R258W}$ mice and human CAPS patients represents an unknown alternative pathway for their maintenance of gut homeostasis. Thus, in our current work, we characterize and explore the possible mechanisms mediating this alternative pathway. We find that $Nlrp3^{R258W}$ mice are strongly resistant to experimental colitis. This is due to an excess of local IL-1β production, but not IL-18, that reshapes intestinal microbiota to induce local $T_{regs}$, to maintain intestinal homeostasis.

## Results

**Gut NLRP3 inflammasome is hyperactive in $Nlrp3^{R258W}$ mice.** To explain why the guts of $Nlrp3^{R258W}$ mutant mice are free of inflammation, a starting question is whether mutated NLRP3 is also expressed and hyperactive in the intestine as in bone marrow-derived macrophages (BMDMs)[21]. Thus, we first examined the expression of NLRP3 in different cell populations in the colon. In isolated intestinal lamina propria mononuclear cells (LPMCs) and epithelial cells (ECs) from control and $Nlrp3^{eGFP}$ reporter mice (in which normal NLRP3 expression is indicated by eGFP expression due to replacement of the $Nlrp3$ coding sequence by eGFP[24]), NLRP3 is expressed in the lamina propria CD11b$^+$/CD11C$^+$ mononuclear phagocytes (LPMPs), but not in other types of cells in the colon (Fig. 1a and Supplementary Fig. 1a). Further subclassification shows that the MHCII$^+$ CD11b$^+$CD11C$^{lo}$CD64$^+$ macrophages represent the largest group of NLRP3-expressing cells, followed by the MHCII$^+$ CD11b$^-$CD11C$^{hi}$ dendritic cells and MHCII$^+$CD11b$^+$CD11C$^{lo}$ CD64$^-$ dendritic cells, as well as MHCII$^-$CD11b$^+$CD11C$^{lo}$Ly6C$^{hi}$ monocytes (Fig. 1a).

Next, we examined whether NLRP3$^{R258W}$ protein maintains its hyperactivity in the gut. Heat-inactivated mouse fecal bacteria (BAC) primed intestinal LPMCs from $Nlrp3^{R258W}$ (+/R258W) mice secrete high levels of IL-1β without ATP pulse, whereas wild-type (WT (+/+)) cells secrete IL-1β only in the presence of ATP. As expected, inflammasome-independent TNF-α production is similar in WT and mutant cells (Fig. 1b, the top and bottom panels). Western blot analysis also confirms the hyperactivity of the NLRP3 inflammasome in NLRP3$^{R258W}$ LPMCs, which is evidenced by the presence of casp1-p10 subunit and mature IL-1β-p17 fragment in the culture supernatant upon BAC stimulation alone, while such signals are detected in WT cells only upon BAC + ATP treatment (Fig. 1c). Of note, neither BAC nor BAC + ATP treatment induces IL-18 secretion in these LPMCs, indicating a discrepant regulation for IL-1β and IL-18 in LPMCs following the activation of the NLRP3 inflammasome (Fig. 1b, middle panel, showing only basal level of IL-18). Therefore, this ATP-independent inflammasome activation and IL-1β induction in the $Nlrp3^{R258W}$ intestinal LPMCs is similar to that in BMDMs as previously reported[21] (also shown in Supplementary Fig. 1b), except that IL-18

induction is not present in LPMCs. In line with previous findings that over-production of IL-1β promotes inflammation by potentiating Th17 cell development and recruiting neutrophils[25,26], elevated numbers of IL-17-generating CD4$^+$ T cells and Ly6G$^{hi}$ neutrophils are observed in the intestines of $Nlrp3^{R258W}$ mice (Supplementary Fig. 1c–e). Taken together, these data demonstrate that the hyperactive NLRP3 inflammasome is competent in the gut of $Nlrp3^{R258W}$ mice, and the IL-1β signaling is also intact, showing an enhanced pro-inflammatory trend locally.

**$Nlrp3^{R258W}$ mice maintain gut homeostasis.** Macroscopically, we found no difference in colon length, number of small intestinal Peyer's patches, or the structure of the intestinal mucosa between the WT and $Nlrp3^{R258W}$ mice (Supplementary Fig. 2a–c). The dynamic intestinal cell proliferation rates are similar between $Nlrp3^{R258W}$ mice and controls (Supplementary Fig. 2d, e). Additionally, the permeability of the intestinal epithelium is comparable between these mice (Supplementary Fig. 2f). Moreover, at tissue level the intestines of $Nlrp3^{R258W}$ mice do not manifest with pan-inflammation, which is evidenced by similar levels of inflammatory mediators, including CCL2, KC, IL-6, IFN-γ, TNF-α, and IL-17A, from the colons of WT and $Nlrp3^{R258W}$ mice, except that IL-12p40 is found significantly lower in the mutated colon tissue (Supplementary Fig. 2g). Interestingly, although $Nlrp3^{R258W}$ mice possess a functional hyperactive inflammasome, at the tissue level the secretion of IL-18 and IL-1β show no difference from that seen in WT mice at a steady state (Supplementary Fig. 2g). Thus, under resting conditions, despite the intrinsic pro-inflammatory propensity, $Nlrp3^{R258W}$ mice maintain intestinal homeostasis.

**$Nlrp3^{R258W}$ mice resist induced colitis and colorectal cancer.** NLRP3 is absent in the epithelium but present in the deeper residing lamina propria. We therefore tested whether the intestines of $Nlrp3^{R258W}$ mice develop more severe inflammation than WT mice following dextran sulfate sodium (DSS)-induced experimental colitis, which would directly breach the epithelial barrier and allow intestinal microbiota-LPMCs confrontation. Surprisingly, $Nlrp3^{R258W}$ mice exhibit significantly less body weight loss, improved survival rates, lower disease scores, and less shortening of colons than WT mice (Fig. 2a–d). Moreover, histological examination of the colons shows that the enteric mucosal structure is better preserved in $Nlrp3^{R258W}$ mice and shows less immune cell infiltration compared to WT controls (Fig. 2e). In addition, colon permeability is significantly lower in $Nlrp3^{R258W}$ than in WT mice after DSS challenge (Fig. 2f). Finally, the colons of the $Nlrp3^{R258W}$ mice are clearly less inflamed, as indicated by the generation of significantly lower levels of inflammatory mediators, including CCL2, KC, IL-6, and CCL5, but the NLRP3 inflammasome downstream signature cytokines, IL-1β and IL-17A, remain high, as does IL-18

**Fig. 2** $Nlrp3^{R258W}$ mice are resistant to DSS-induced colitis and colorectal cancer. **a–c** Colitis was induced in the WT ($n = 32$) and $Nlrp3^{R258W}$ ($n = 21$) mice using 2.5% DSS. The body weights (**a** change in percentage), survival rates (**b** change in percentage), and disease scores (**c** arbitrary unit) were assessed daily. **d** The colon lengths were measured from the cecum–colon junction to the anus end of a loosely stretched colon with a straight ruler upon killing of indicated experimental mice on day 8 and representative examples are shown. **e** Hematoxylin and eosin (H&E)-stained transverse sections of the colons were examined and scored as described in the "Methods" section. Representative images of these samples are shown; the scale bar is 200 μm. **f** WT ($n = 6$ for 3 days, $n = 3$ for 5 days) and $Nlrp3^{R258W}$ ($n = 3$ for 3 days, $n = 3$ for 5 days) mice were treated with 2.5% DSS in drinking water for indicated days, then they were gavaged with FITC-Dextran and leakage of FITC-Dextran into the serum was measured 4 h later. **g** The indicated cytokines or chemokines in the culture supernatants of colon tissues from mice as shown in **d** were determined via ELISA. **h** The indicated transcripts from colon tissue homogenates from mice as shown in **d** were analyzed by Q-PCR. AU arbitrary unit. **i–l** WT ($n = 10$) and $Nlrp3^{R258W}$ ($n = 11$) mice were treated with AOM + DSS to induce colon cancer. The body weights (**i**) and survival rates (**j**) were monitored. The final tumor loads (**k**) were shown as representative images, and the tumor size distributions (**l**) were determined. The data are shown as the means ± SEM. *$P < 0.05$, **$P < 0.01$, ***$P < 0.001$ (Two-tailed Student's $t$ test). Survival rate **b**, **j** difference is tested by Log-rank (Mantel–Cox) test, *$P < 0.05$

produced by stimulated epithelial cells (see "Discussion" section) in mutated colon (Fig. 2g, h). These results suggest that mice carrying the *Nlrp3*[R258W] mutation develop significantly fewer gut disease than WT controls in response to DSS treatment.

Unresolved and recurrent colitis is a risk for colorectal cancer. Thus, we next examined whether NLRP3[R258W] confers protection against colorectal cancer using a mouse model of chronic azoxymethane (AOM) + DSS-induced colitis instigated colorectal cancer. Indeed, compared with *Nlrp3*[R258W] mice, colitis severity in the first two cycles is significantly higher in WT mice, as reflected by the weight loss and the early deaths of WT mice during the second cycle of DSS treatment (Fig. 2i, j). Additionally,

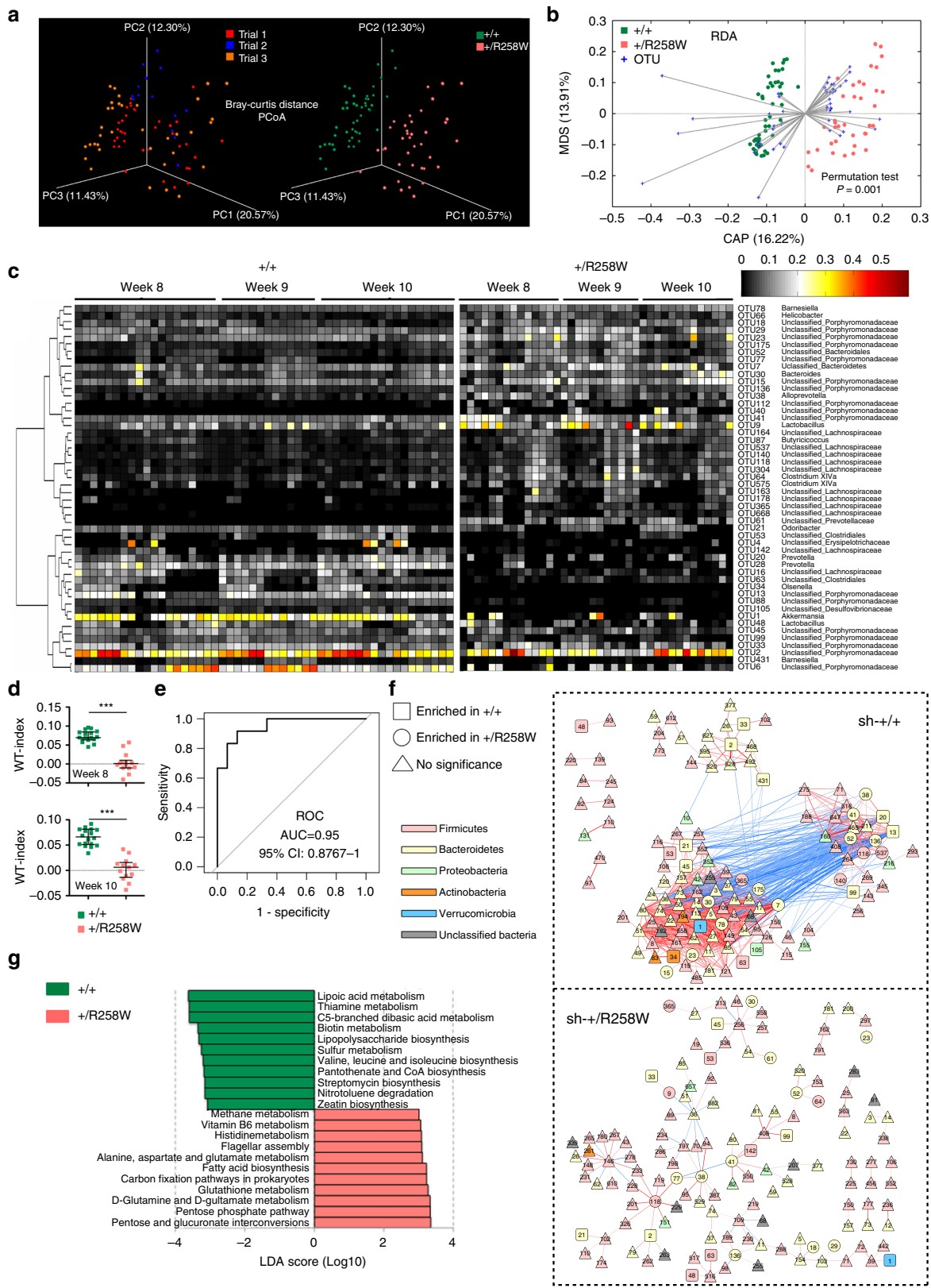

at the end point of this experiment, $Nlrp3^{R258W}$ mice have significantly fewer and smaller tumors in the colon than WT mice (Fig. 2k, l). Therefore, $Nlrp3^{R258W}$ mice not only maintain intestinal homeostasis under resting conditions, but also have strong resistance to chemical-induced colitis and colorectal cancer.

**$Nlrp3^{R258W}$ mice have unique gut microbiota**. In the balance between intestinal microbiota and the host immune system, deviations on either side may cause corresponding adjustment in the other. We thus hypothesized that the ability to maintain homeostasis under resting conditions and the resistance against experimental colitis and colorectal cancer in $Nlrp3^{R258W}$ mice might be attributable to an alteration in the gut microbiota. To test this, we sequenced the 16S rRNA gene variable (V) 3-V4 region of the fecal bacteria obtained from three independent experiments (marked as trials 1, 2 and 3 in Fig. 3a). Although the richness and diversity of the intestinal microbiota do not differ (Supplementary Fig. 3a), the overall structure of the gut microbiota shows a striking difference between WT and $Nlrp3^{R258W}$ mice (Fig. 3a). At the phylum level, the abundance of dominant intestinal bacteria such as Bacteroidetes and Firmicutes are similar between these two genotypes, but Actinobacteria and Verrucomicrobia are significantly less abundant in $Nlrp3^{R258W}$ mice (Mann–Whitney test, FDR < 0.05, Supplementary Fig. 3b). At the phylotype level, through redundancy analysis (RDA) using the abundance of operational taxonomic units (OTUs, ≥97% similarity), the microbiota from WT and $Nlrp3^{R258W}$ mice is significantly different, in that 30 OTUs (including *Lactobacillus* OTU9) are more abundant and 20 OTUs (including *Akkermansia* OTU1) are less abundant in $Nlrp3^{R258W}$ than in WT mice (Fig. 3b, c and the full list is shown in Supplementary Table 1). Based on these 50 OTUs, we defined a WT-index, which is a measurement of microbiota along the axis representing the major difference between WT and $Nlrp3^{R258W}$ mice (see "Methods" section). The WT-index values for WT and $Nlrp3^{R258W}$ mice microbiota are significantly different (Fig. 3d). We used microbiota from an additional trial (Trial 4) to validate the WT-index model. The area under the receiver operating characteristic (ROC) curve is 0.95 (leave-one-out cross-validation, 95% confidence interval (0.88, 1.00), Fig. 3e), indicating that the WT-index accurately defines the differences between WT and $Nlrp3^{R258W}$ gut microbiota. Thus, hereafter we used this WT-index as a scale to measure whether a microbiota is more similar to WT or to $Nlrp3^{R258W}$ mice.

In addition to microbiota composition difference, we also found that during DSS challenge, the microbial co-abundance network at the OTU level in $Nlrp3^{R258W}$ mice was notably different from that in WT controls (Fig. 3f). Specifically, the network in $Nlrp3^{R258W}$ mice has more isolated nodes (OTUs), longer characteristic path length, lower clustering coefficiency, reduced network centralization, less average number of

neighbors, and lower network density than in WT mice (Supplementary Table 2). The network degree and eigenvector centrality, a measure of the relative importance and the connectivity of each node in a network[27], are also significantly lower in $Nlrp3^{R258W}$ mice (Kolmogorov–Smirnov tests, $P < 0.01$, Supplementary Fig. 4a), implying that the information flow varies throughout the gut microbial networks between the two genotypes. Moreover, the gut microbiota of $Nlrp3^{R258W}$ mice show higher deviation from baseline than that of WT mice on the first 3 days of DSS challenge (Supplementary Fig. 4b). Together, these data reveal that the gut bacteria-interacting network in WT mice is denser with a higher degree distribution per node, whereas the $Nlrp3^{R258W}$ mutation significantly reduces the interactions in the gut microbial community, making it more fluid.

To see how the altered community structure of the gut microbiota affects its function, we also performed Illumina-based shotgun metagenomic sequencing of fecal bacteria DNA. The PCA score plot of all the KEGG orthology groups (KOs) shows a significant separation between WT and $Nlrp3^{R258W}$ mice (Supplementary Fig. 4c). Using the linear discriminant analysis (LDA) effect size (LEfSe) method[28], 11 KEGG database biochemical pathways are significantly decreased and 11 are enriched in $Nlrp3^{R258W}$ mice (Fig. 3g and Supplementary Fig. 4d). Notably, some pathogen-associated pathways, such as biosynthesis of lipopolysaccharide and lipoate metabolism[29, 30], are decreased while glutathione metabolism and D-glutamine/D-glutamate metabolism, which were interpreted as antioxidant pathways improving host epithelium integrity[31], are increased in $Nlrp3^{R258W}$ microbiota. These data suggest a significantly less pro-inflammatory signature, as well as an increased anti-inflammatory capacity of the gut microbiome in $Nlrp3^{R258W}$ mice. Taken together, the $Nlrp3^{R258W}$ mutation reshapes gut microbiota with a distinct composition, network topology, and functionality.

**Reshaped $Nlrp3^{R258W}$ microbiota confer disease resistance**. Next, we addressed whether the alteration of $Nlrp3^{R258W}$ microbiota directly underlies the increased resistance to colitis/colorectal cancer. To test this, we determined the susceptibility to colitis/colorectal cancer in $Nlrp3^{R258W}$ and WT mice under various conditions in which the gut microbiome has been altered.

We first cohoused WT and $Nlrp3^{R258W}$ mice at a 1:1 ratio from weaning to adulthood to exchange their microbiota, and then induced colitis with DSS. We found that the overall structure of the fecal microbiota from WT and $Nlrp3^{R258W}$ mice was mutually affected during cohousing, but shifted more toward that of the WT animals (Fig. 4a–d). This mutual shifting pattern is also evident in the WT-index (Fig. 4e) and the microbial network measurements (Supplementary Table 2 and Fig. 4f). Moreover, the network degree and eigenvector centrality distributions after cohousing are more similar to WT than to $Nlrp3^{R258W}$ mice

**Fig. 3** $Nlrp3^{R258W}$ mice carry a unique gut microbiota with a sparingly inerconnected ecology and altered functionality. **a–f** Fecal microbiota composition was assayed by bacterial 16S rRNA V3–V4 region sequencing: **a** PCoA plot (based on Bray–Curtis distance) of the gut microbiota from three trials of WT ($n = 18$, 9, and 18) and $Nlrp3^{R258W}$ ($n = 18$, 5, and 15) mice. **b** Redundancy analysis ordination biplot based on the Hellinger-transformed Euclidean distance among samples and the eigenvalues for 50 OTUs (which explained more than 10% of the variability of the samples). **c** Heat map of the 50 key OTUs in gut microbiota from WT and $Nlrp3^{R258W}$ mice at the age of 8, 9, and 10 weeks in trials 1–3. **d** The WT indices of the gut microbiota from WT and $Nlrp3^{R258W}$ mice at the age of 8 and 10 weeks in trials 1–3. The medians with interquartile ranges are shown. ***$P < 0.001$ (Mann–Whitney test). **e** The area under the ROC curve (AUC) of the gut–microbiota-based classification of the WT and $Nlrp3^{R258W}$ mice. The WT-index was computed for another set of WT ($n = 12$) and $Nlrp3^{R258W}$ ($n = 15$) samples (trial 4). **f** The network of OTUs (shared by more than 40% of samples) in the WT (+/+) and $Nlrp3^{R258W}$ (+/R258W) mice during DSS treatment ($n = 6$ from day −20 to day 2 and $n = 4$ on days 3 and 8). The lines represent positive (red) and negative (blue) correlations between the nodes, with the line width indicating the correlation magnitude. Only nodes with |R| > 0.7 are drawn, and unconnected nodes have been omitted. **g** Metagenomic analysis of key pathways of gut microbiota shift in $Nlrp3^{R258W}$ mice. Histogram shows the LDA scores computed for features (on the pathway level) that are differentially abundant between the WT and $Nlrp3^{R258W}$ mice

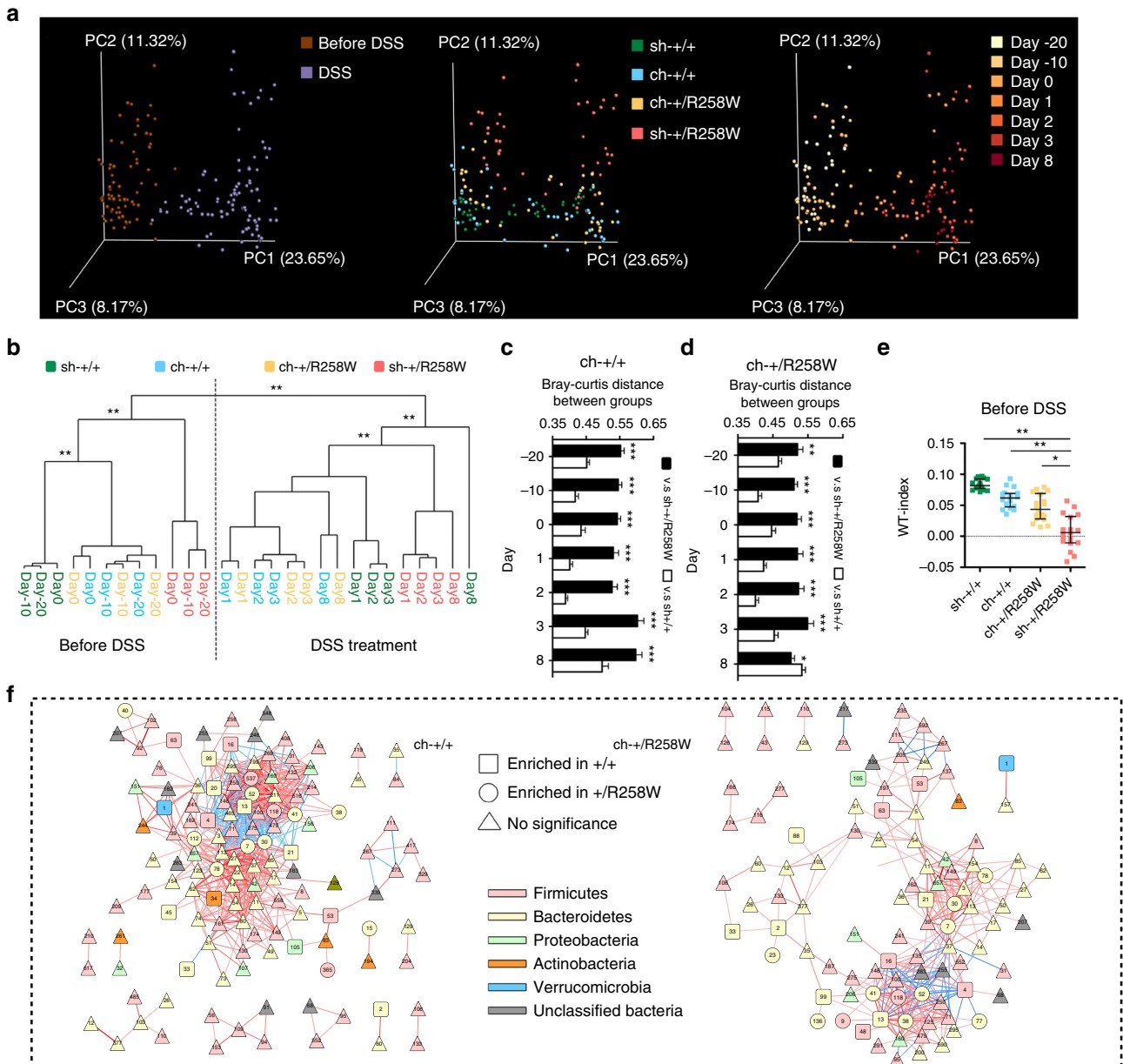

**Fig. 4** The gut microbiota of the *Nlrp3*[R258W] mice is affected by cohousing with WT littermates. **a–f** Fecal microbiota composition was assayed by bacterial 16S rRNA V3–V4 region sequencing: **a** PCoA plot (based on Bray–Curtis distance) of the fecal microbiota of singly housed WT (sh-+/+), cohoused WT (ch-+/+), cohoused *Nlrp3*[R258W] (ch-+/R258W), and singly housed *Nlrp3*[R258W] (sh-+/R258W) mice (*n* = 6 in each group) before and during DSS treatment. **b** Clustering of the gut microbiota based on distances between different groups with the first 16 PCs (accounting for 80% of total variations) of the PCoA based on the Bray–Curtis distance. **P < 0.01 (MANOVA test). (**c**, **d** The Bray–Curtis distance of the ch-+/+ (**c**) and ch-+/R258W (**d**) gut microbiota to the sh-animals was calculated. The data are shown as means ± SEM. *P < 0.05, **P < 0.01, ***P < 0.001 (two-tailed Student's *t* test). **e** The WT indices of sh-+/+, ch-+/+, ch-+/R258W, and sh-+/R258W mice before DSS treatment: the medians with interquartile ranges are shown. *P < 0.05, **P < 0.01 (Kruskal–Wallis test). **f** The network of OTUs (shared by more than 40% samples) in co-house WT mice and co-house *Nlrp3*[R258W] mice during DSS treatment (*n* = 6 from day −20 to day 3 and *n* = 4 on day 8). The red lines represent positive correlations between the nodes and the blue lines represent negative correlations, with line width indicating the correlation magnitude. Only the IRI is greater than 0.7 are drawn, and unconnected nodes have been omitted (refer to Fig. 3f)

(Supplementary Fig. 4a). Concomitantly with the gut microbiome alteration, the disease differences between these two lines of mice also reduced, as demonstrated by the convergent weight loss, colon lengths, and disease scores (Fig. 5a–c and Supplementary Fig. 5a). Additionally, cohousing exacerbated the histological pathology in *Nlrp3*[R258W] mice, whereas the pathology in WT mice hardly improved (Fig. 5d, e). Moreover, the production of IL-6 and CCL2, as well as bacterial dissemination to the mesenteric lymph nodes (MLN) and livers also fall in between

those of the singly housed animals (Supplementary Fig. 5b, c); while the signature cytokines following NLRP3 inflammasome activation, e.g., IL-1β and IL-17A, are still higher in the mutated tissue (Supplementary Fig. 5b). We noticed that unlike IL-1β and IL-17A, which seem to be rather stringently dependent on the genotype, IL-18 displays a strongly reversed correlation with disease severity (noted in "Discussion" section) (Supplementary Fig. 5b). Finally, colon cancer progression and tumor burdens (Fig. 5f, g and Supplementary Fig. 5d) also display mutual shifting

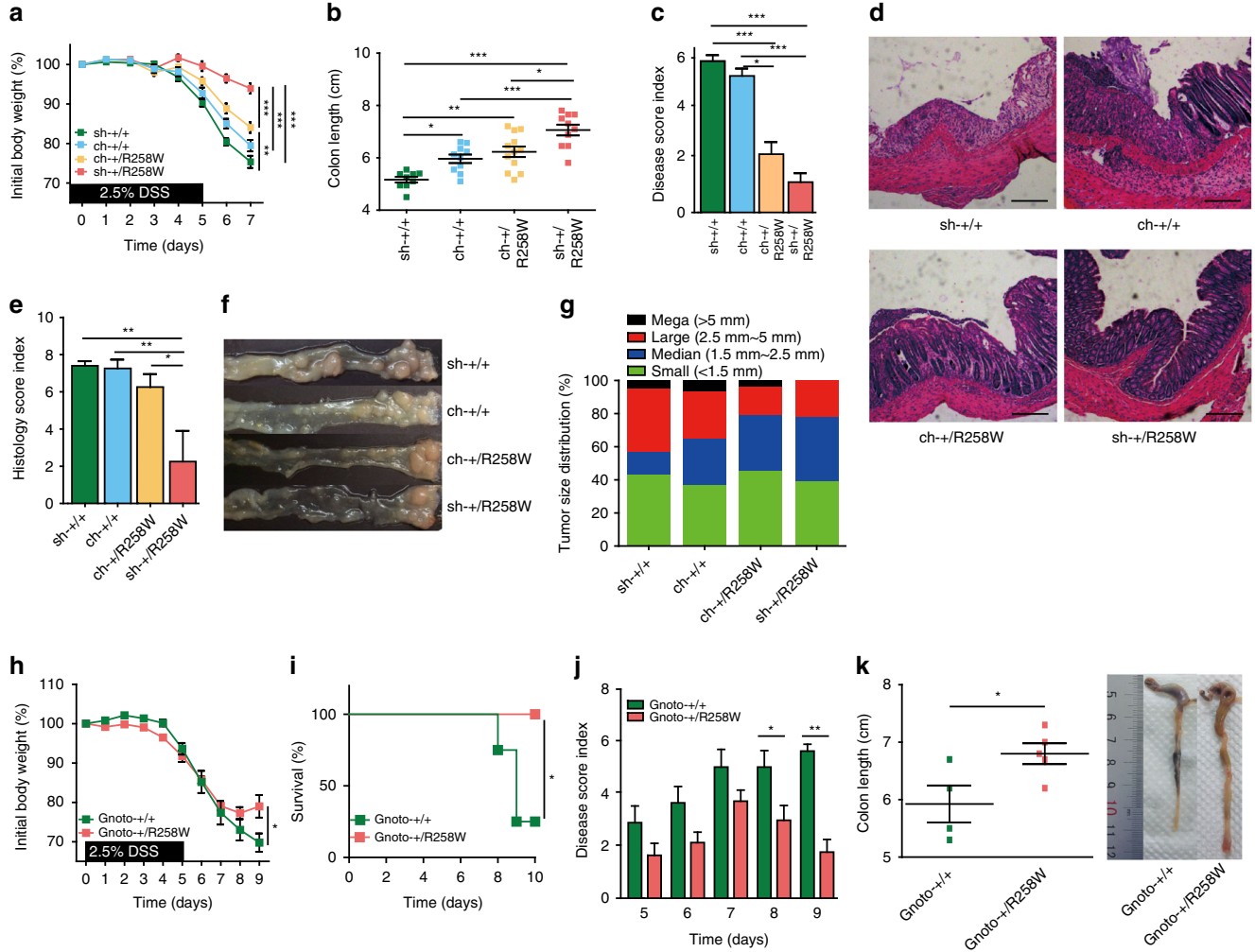

**Fig. 5** The gut microbiota of *Nlrp3*<sup>R258W</sup> mice confers resistance to DSS-induced colitis and colorectal cancer. **a**–**e** Acute colitis was induced in sh-+/+ (*n* = 13) and sh-+/R258W (*n* = 10) mice and 1:1 cohoused (ch-, *n* = 12 for each genotype) mice. The analyses of body weights (**a**), colon length (**b**), disease scores (**c**), histological examination (**d**), scale bar is 200 μm, and histological score (**e**) were conducted as described in Fig. 2, except that the mice were killed on day 7. The data in **a**, **b**, **c** and **e** are shown as the means ± SEM. *P < 0.05, **P < 0.01, ***P < 0.001 (one-way ANOVA with Tukey's post-hoc test). **f**, **g** Colon cancer was induced and monitored in singly housed and cohoused mice (*n* = 6 for each group), the representative colon tumor (**f**) and total tumor size distribution (**g**) were shown. **h**–**k** Feces from the WT and *Nlrp3*<sup>R258W</sup> mice were transplanted to germ-free WT mice (Gnoto-+/+, *n* = 4, Gnoto-+/R258W, *n* = 5), and acute colitis was then induced with 2.5% DSS. The diseases were assayed as described in Fig. 2, except that the mice were killed on day 9. The data in **h**, **j** and **k** are shown as the means ± SEM. *P < 0.05, **P < 0.01 (two-tailed Student's *t* test). Survival rate is shown in **i**, *P < 0.05 (Log-rank (Mantel–Cox) test)

after cohousing. Thus, manipulation of the composition and ecology of the gut microbiota through cohousing leads to corresponding changes in colon pathology in experimental colitis and colorectal cancer.

In further studies assessing the effects of the altered gut microbiome on colitis resistance in *Nlrp3*<sup>R258W</sup> mice, we depleted the gut microbiota with antibiotics (see "Methods"). In this case, DSS induces only very mild colitis, and the disease difference between *Nlrp3*<sup>R258W</sup> and WT mice disappeared (Supplementary Fig. 5e). Next, we transferred feces from either WT or *Nlrp3*<sup>R258W</sup> mice housed in a specific pathogen-free facility to WT germ-free mice (denoted as Gnoto-+/+ or Gnoto-+/R258W, respectively) via oral gavage. After colonization, the microbiota from the WT and the *Nlrp3*<sup>R258W</sup> mice maintain their original state (Supplementary Fig. 5f). After DSS challenge, the Gnoto-+/R258W mice show significantly less disease progression, better survival rates, lower disease scores, and less shortening of colons than the Gnoto-+/+ mice (Fig. 5h–k). These data together with the data from the cohousing studies verify that resistance to colitis and

colon cancer in the *Nlrp3*<sup>R258W</sup> mice is mainly mediated by the altered gut microbiota that are reshaped by the genetic mutation.

**IL-1β-induced antimicrobial peptides reshape microbiota.** After we confirmed that the hyperactive NLRP3 inflammasome is competent exclusively in the CD11b<sup>+</sup>/CD11C<sup>+</sup> LPMPs of *Nlrp3*<sup>R258W</sup> mice intestines, we looked at how this mutation reshapes the gut microbiota. Following activation of the NLRP3 inflammasome, IL-1β and IL-18 are processed as effector molecules linking subsequent signaling. As noted in Fig. 1b, different from IL-1β, IL-18 secretion is absent in both WT and *Nlrp3*<sup>R258W</sup> colon LPMCs after NLRP3 inflammasome activation. Thus, we examined the expression of these cytokines in the intestinal microenvironment under resting status. Interestingly, little pro-IL-18 is expressed in intestinal LPMCs, which clearly express NLRP3 and pro-IL-1β (from both WT and *Nlrp3*<sup>R258W</sup> mice), while in intestinal epithelial cells (ECs), which express abundant pro-IL-18, hardly any NLRP3 or pro-IL-1β is detected (Fig. 6a, b

and Supplementary Fig. 6a). Indeed, in epithelial cells where no NLRP3 is expressed, a clear IL-18 signal is detected, but it is equivalent in WT and *Nlrp3*[R258W] mice (Supplementary Fig. 6b). These findings indicate that IL-1β, but not IL-18, is the most likely effector molecule directly downstream of the NLRP3 inflammasome in the intestine. To functionally validate this unique feature for NLRP3 inflammasome in the gut, we isolated both LPMCs and ECs from untreated WT, *Nlrp3*[−/−] and

*Caspase1/11*[−/−] mice and measured the IL-1β and IL-18 production from these cells. We found that IL-1β is secreted by LPMCs but not ECs under BAC + ATP stimulation, and it is stringently dependent on both NLRP3 and caspase-1 (Fig. 6c, top panel). On the contrary, IL-18 is only secreted by ECs, and is dependent on neither BAC + ATP stimulation nor NLRP3, but on caspase-1 (Fig. 6c, bottom panel). Previous reports claimed that in the DSS colitis and colorectal cancer models, IL-18 is crucial

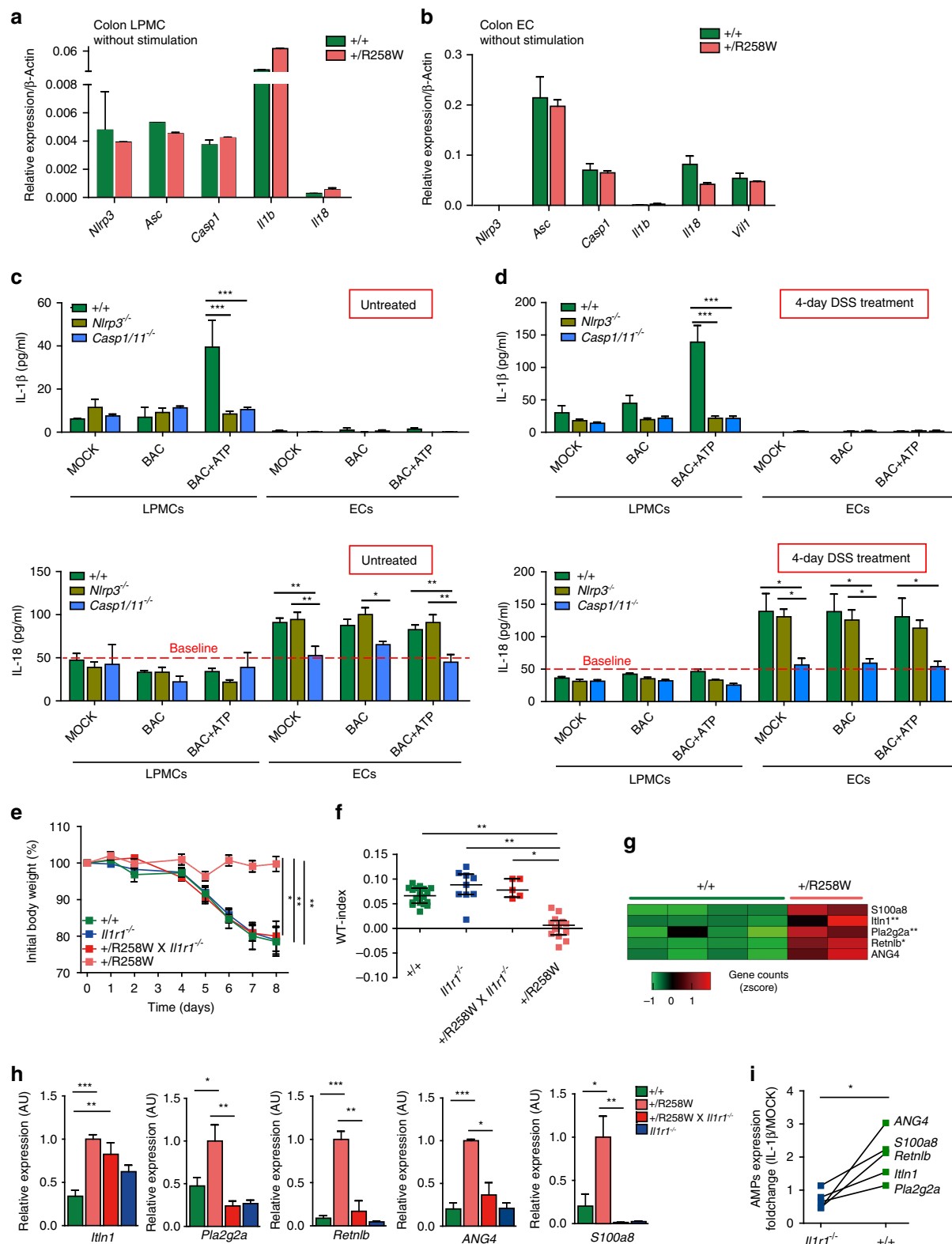

for NLRP3 inflammasome-mediated protection[32, 33]. So we next tested under the DSS challenging condition whether IL-18 production is dependent on NLRP3 inflammasome. Interestingly, after 4-day DSS treatment, the secretion pattern of neither IL-1β nor IL-18 changed, but the production level for both cytokines increased compared to cells from naive mice (Fig. 6d). Together with the mRNA expression profile, these results further validate a functional role for IL-1β but not IL-18 directly downstream of NLRP3 in the intestinal LPMCs; IL-18 might be indirectly affected by NLRP3, if at all, through secondary effects. Furthermore, when $Nlrp3^{R258W}$ mice were crossed to an $Il1r1^{-/-}$ background to abolish the IL-1β signaling, these mice regained susceptibility to DSS-induced colitis (Fig. 6e and Supplementary Fig. 6c). Concomitantly, the intestinal microbiota of these mice is more similar to that of WT mice (Fig. 6f and Supplementary Fig. 6d). These results thus confirm that the $Nlrp3^{R258W}$ mutation shapes the microbiota through IL-1β. Interestingly, we found that IL-18 deficiency can also abolish the protective effect of NLRP3$^{R258W}$ (Supplementary Fig. 6e, f). However, our cellular analysis did not find a direct connection between NLRP3 and IL-18 in the intestine. The observation that $Nlrp3^{R258W} \times Il18^{-/-}$ mice lose resistance to DSS-induced colitis is probably due to the more predominant and frontier role of IL-18 in the epithelium than NLRP3 in the lamina propria.

IL-1β has been shown to play an important role in regulating the production of antimicrobial peptides (AMPs)[11]. We therefore asked whether the $Nlrp3^{R258W}$ mutation also regulates AMPs expression in the gut. Indeed, in our RNAseq analysis of gene expression differences between $Nlrp3^{R258W}$ and WT colon tissues, we find three specific AMPs, Itln1, sPLA2, and Retnlb, that are significantly upregulated in the colons of $Nlrp3^{R258W}$ mice (Fig. 6g). Quantitative PCR (Q-PCR) analysis validated these genes together with other two AMPs, Angiogenin4 (ANG4) and S100A8, which show similar trends (Fig. 6h). More importantly, we find that the upregulation of these AMPs is dependent on IL-1 signaling, as absence of IL-1 signal on an $Il1r1^{-/-}$ background, abolishes the expression of these genes except Itln1 (Fig. 6h). Ex vivo stimulation of colon tissue with recombinant IL-1β also directly increases the expression of these AMPs to varied extent (Fig. 6i). Taken together, these results demonstrate that the NLRP3$^{R258W}$ protein acts in the CD11b$^+$/CD11C$^+$ LPMPs to promote gut homeostasis, at least in part, through boosting the local production of AMPs by IL-1β.

**Microbiota upregulate T$_{regs}$ to neutralize inflammation**. Our data to this point suggest that modulation of the gut microbiota by the $Nlrp3^{R258W}$ mutation leads to increased resistance to gut inflammation. We then looked at the possible mechanism for this. Given the essential role for Foxp3$^+$ T regulatory cells (T$_{regs}$) in maintaining homeostasis and combating inflammatory

conditions in the gut, and the ability of certain microbial species to promote T$_{regs}$ differentiation[34, 35], we looked at whether the microbiota in $Nlrp3^{R258W}$ mice also upregulate T$_{regs}$ to neutralize inflammation.

We enumerated Foxp3$^+$ T$_{regs}$ in the colonic lamina propria of $Nlrp3^{R258W}$ and WT mice. Indeed, these cells are significantly more numerous in $Nlrp3^{R258W}$ mice (Fig. 7a). Moreover, through cohousing of these mice, we find that the upregulation of T$_{regs}$ is dependent on the microbiota (Fig. 7a), and that T$_{regs}$ abundance in the colon correlates well with colitis resistance (Fig. 5a–e). To validate the T$_{reg}$ induction by the reshaped microbiota in $Nlrp3^{R258W}$ mice, we transferred WT and $Nlrp3^{R258W}$ fecal microbiota to germ-free recipient mice via oral gavage. We found that $Nlrp3^{R258W}$ microbiota is significantly more capable of inducing T$_{regs}$ in the colon (Fig. 7b). Since the microbiota is shaped by IL-1β signaling, we examined the T$_{regs}$ in $Nlrp3^{R258W}$ mice on the $Il1r1^{-/-}$ background and found that T$_{regs}$ in these mice were reduced to those seen in WT (Fig. 7c); this also correlated with severity of colitis (Fig. 6e) and so suggests a crucial role for T$_{regs}$ in microbiota-mediated protection of mice from colitis. Next, we sought to inhibit T$_{regs}$ in $Nlrp3^{R258W}$ mice and then induce DSS colitis to see if the protective effects are compromised. Administration of anti-CD25-neutralizing antibody has been reported as a reliable way to reduce T$_{regs}$ locally and systematically[36]. Thus, we i.p. injected anti-CD25-neutralizing antibody in $Nlrp3^{R258W}$ mice followed with colitis induction. The antibody-mediated depletion of T$_{regs}$ in mutant mice was confirmed (Supplementary Fig. 7a). Indeed, the colitis severity in T$_{regs}$-depleted $Nlrp3^{R258W}$ mice increased significantly compared to $Nlrp3^{R258W}$ mice receiving isotype antibody, as shown by the weight loss, clinic scores and colon length, which were similar to WT control mice (Fig. 7d, e and Supplementary Fig. 7b, c).

To further examine the role of T$_{regs}$ in $Nlrp3^{R258W}$ mice, we tried to examine the protective effect of these cells in a T cell transfer colitis model. We reasoned that in such a model the induction of colitis with effector T cells without the presence of T$_{regs}$ in $Nlrp3^{R258W}$ x $Rag1^{-/-}$ mice could abrogate the protective function of remodelled microbiota, and lead to greater severity of disease given that the mutation is pro-inflammatory (Fig. 1 and supplementary Fig. 1). Surprisingly, we found that after crossing to the $Rag1^{-/-}$ background, $Nlrp3^{R258W}$ mice develop notable spontaneous colitis (Fig. 8a). Compared with $Rag1^{-/-}$ and $Nlrp3^{R258W}$ mice, $Nlrp3^{R258W}$ mice on a $Rag1^{-/-}$ background show severely inflamed colons and ileums with significantly shortened colon length and obvious diarrhea (Fig. 8a–c). Histological examination shows that these mice have extensive immune cell infiltration and severe deformation of the mucosa in their intestines (Fig. 8d, e). Additionally, a panel of inflammatory mediators, including CCL2, KC, IL-1β, IL-17A, and TNF-α, are significantly higher only in the $Nlrp3^{R258W}$ x $Rag1^{-/-}$ mice, while

**Fig. 6** The $Nlrp3^{R258W}$ gut microbiota is shaped by IL-1β-induced antimicrobial peptides. **a**, **b** Colonic LPMCs and epithelial cells (ECs) were isolated from WT and $Nlrp3^{R258W}$ mice and assayed for the expression of the indicated genes via Q-PCR without stimulation. The data are shown as the means ± SEM. **c**, **d** Colonic LPMCs and ECs were isolated from naive (**c** n = 3 for each genotype) or 4 days 4% DSS-treated (**d** n = 4 for each genotype) WT, $Nlrp3^{-/-}$ and $Casp1/11^{-/-}$ mice, the cells were treated with BAC (MOI = 20:1) ± ATP, and the culture media was assayed for IL-1β and IL-18 secretion via ELISA. The data are shown as the means ± SEM. *P < 0.05, **P < 0.01, ***P < 0.001 (one-way ANOVA with Tukey's post-hoc test). **e** Changes in body weight in WT (n = 8), $Il1r1^{-/-}$ (n = 9), $Nlrp3^{R258W}$ x $Il1r1^{-/-}$ (n = 6), and $Nlrp3^{R258W}$ (n = 5) mice during DSS-induced colitis. The data are shown as the means ± SEM, *P < 0.05, **P < 0.01 (one-way ANOVA with Tukey's post-hoc test). **f** The microbiota WT-index was also calculated for mice in **e** before DSS treatment. The medians with interquartile ranges are shown. *P < 0.05, **P < 0.01 (Kruskal–Wallis test). **g** Untreated WT (n = 4) and $Nlrp3^{R258W}$ (n = 2) mice colon tissue RNA was assayed by RNAseq analysis, selected AMPs were shown to be upregulated in the $Nlrp3^{R258W}$ colon at varied extent. *P < 0.05, **P < 0.01 (statistical comparison was conducted using DESeq2, see "Methods" section). **h** AMPs expression in untreated colon tissue from WT, $Nlrp3^{R258W}$, $Nlrp3^{R258W}$ x $IL1r1^{-/-}$ and $Il1r1^{-/-}$ mice were analyzed by Q-PCR. AU arbitrary unit. The data are shown as the means ± SEM, *P < 0.05, **P < 0.01, ***P < 0.001 (one-way ANOVA with Tukey's post-hoc test). **i** Colon tissues from $Il1r1^{-/-}$ and WT mice were stimulated with 50 ng/ml mIL-1β for 4 h. Fold changes of AMPs expression (IL-1β stimulation vs. untreated) in colon tissues are shown. *P < 0.05 (two-tailed paired t test)

the level of IL-18 does not show a clear difference, again arguing that IL-18 is not directly connected with $Nlrp3^{R258W}$ in the gut (Fig. 8f, g). These data provide strong evidence to support that T$_{regs}$ are critical for restricting inflammation in the gut of $Nlrp3^{R258W}$ mice. They also suggest that the $Nlrp3^{R258W}$ mutation in the intestine is intrinsically pro-inflammatory, but the local microbiota helps host maintain homeostasis through an alternative pathway using T$_{regs}$.

Collectively, our data show that the microbiota is reshaped by increased IL-1 signaling in $Nlrp3^{R258W}$ mice, which supports the development of local T$_{reg}$ cells that directly neutralize gut inflammation.

## Discussion

Using the auto-inflammatory $Nlrp3^{R258W}$ mutant mouse line, our current study reveals an unexpected scenario where a hyperactive NLRP3 inflammasome improves the gut symbiosis through microbiota increasing T$_{regs}$ induction. This not only maintains gut homeostasis, but also confers strong resistance to

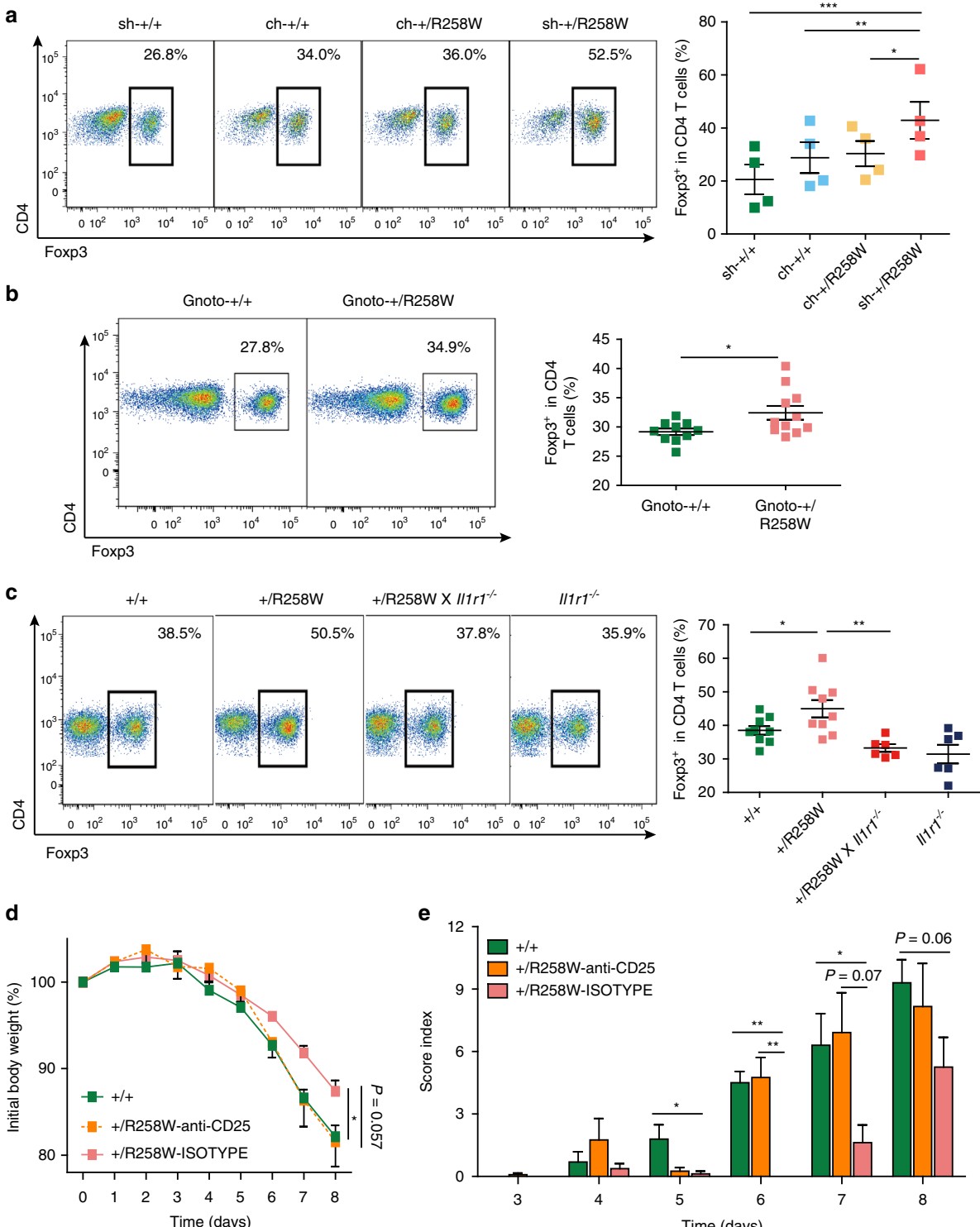

experimental colitis/colorectal cancer. Our findings thus underscore the crucial contribution of the plasticity of microbiota as countermeasures to genetic defects in promoting the cooperative host–microbiota mutualism.

Multiple studies have suggested an involvement of NLRP3 in experimental colitis and colorectal cancer[37], but the cellular location of NLRP3 in the gut under steady state has not been well characterized. Our study demonstrates that the NLRP3 protein is exclusively expressed in CD11b$^+$/CD11C$^+$ LPMPs in the intestine at resting stage. The mutated NLRP3$^{R258W}$ in these cells is hyperactive, leading to local over-production of IL-1β but not IL-18. To our knowledge, this is the first study to identify such a mechanism for NLRP3 inflammasome in the intestine. Our data coincide with a recent study, which also shows that NLRP3 and IL-1β are expressed selectively in the lamina propria, while IL-18 is expressed only in the gut epithelium both in mice and humans[38]. In a separate study, the epithelium caspase-1 activation and IL-18 secretion were found to be independent of NLRP3, which also confirms our findings[39]. We also found that 4 days after DSS challenge, IL-18 is hardly detectable in the lamina propria, whereas a significant increase is observed in epithelial cells in a NLRP3-independent manner. These data indicate that at the early stage of an inflammatory condition in the gut, NLRP3 may still only engage IL-1β for downstream signaling. Our observation that IL-18 is higher in the Nlrp3$^{R258W}$ mice colon after DSS colitis (Fig. 2g and Supplementary Fig. 5b) is likely due to there being less damage to the epithelium in these mice (Figs. 2e, f and 5d), which allows more efficient generation of IL-18 from the epithelium under challenge. But during chronic colitis or colorectal cancer, NLRP3 in infiltrated myeloid cells might also contribute to IL-18 production[33, 40]. Thus, in our model, it is the elevated IL-1β production from Nlrp3$^{R258W}$ mutant mice that enhances the secretion of specific AMPs in the colon, which promotes the development of an anti-inflammatory microbiota.

Of note, the Nlrp3$^{R258W}$ mutation-altered microbiota confer resistance to colitis after being transplanted into germ-free mice, indicating that the anti-inflammatory capacity of the newly established microbiota structure can function independently from the host's genetic background. The distinct microbiota in Nlrp3$^{R258W}$ mice promote local differentiation of T$_{regs}$, which contribute to maintaining homeostasis in the intestine and resistance against gut inflammation (Fig. 7). Among the species (OTUs) enriched in the microbiota of Nlrp3$^{R258W}$ mice, we observed the proliferation of some OTUs, including Clostridium XIVa (OTU64 and OTU575) and Lactobacillus murinus (OTU9), that are closely related to species or genera previously reported to enhance the T$_{regs}$ population[35, 41–43]. Thus, the anti-inflammatory capacity of the Nlrp3$^{R258W}$ altered microbiota may be dependent on the enrichment of T$_{regs}$-promoting bacteria. In addition, we found that several OTUs (OTU1 Akkermensia muciniphila, OTU20 and OTU28 of the genus Prevotella), which

have been shown to be colitogenic[10, 44], are significantly reduced in Nlrp3$^{R258W}$ mice microbiota. Reduction of these species, together with the reduced pathogen-associated pathways identified by metagenomic analysis, suggests an overall intrinsically less pathogenic/pro-inflammatory nature of Nlrp3$^{R258W}$ mice microbiota. As a causal relationship between this gut microbiota structure and colitis susceptibility has been proved in our mouse model, it implies a novel criterion for diagnosis or treatment of inflammatory bowel diseases (IBD) in the future.

The changes in the microbiota topologies of microbial interaction networks have been reported to be associated with alcoholic liver disease and type 1 diabetes[45, 46], but the contribution of such changes in microbiota topology to IBD has not been studied. Our current study demonstrates that the gut microbiota in Nlrp3$^{R258W}$ mice is remodelled not only in the composition, but also in the topology of the microbial co-abundance network in response to DSS challenge. The significantly denser network of microbiota in the WT mice suggests a more stable micro-ecological community[47], which is at least partially confirmed by its slower changes during DSS treatment and more invasiveness against Nlrp3$^{R258W}$ microbiota during cohousing. However, the more fluid gut microbiota in Nlrp3$^{R258W}$ mice has fewer pathogen-associated pathways and a greater ability to attenuate oxidative stress, which may help the host mitigate inflammatory damages. Thus, in addition to its composition, our finding unveils a previously unappreciated aspect (topology) of gut microbiota relating to IBD. But whether this topology of microbiota network represents a universal causal factor underlying the anti-inflammatory capacity of a given microbiota irrespective of its composition needs to be further studied.

Probably because the microbiota composition had not been monitored, several earlier studies on the role of NLRP3[12, 48–50] and IL-1β[51, 52] in experimental colitis or colon cancer using the gene-deficient mice reported discrepant results, leaving the role for NLRP3 inflammasome in inflammatory bowel disease in ambiguity for the time. Thus, our study using the gain-of-function Nlrp3$^{R258W}$ mice and monitoring their microbiota shift has dissected a complex crosstalk between NLRP3 inflammasome and indigenous microbiota to contribute to gut homeostasis, emphasizing microbiota's crucial role in mediating NLRP3 inflammasome function in the intestine, providing critical insights in this area.

The observation that Nlrp3$^{R258W}$x Rag1$^{-/-}$ mice develop spontaneous colitis strongly supports the hypothesis that the Nlrp3$^{R258W}$ mutation in the intestine is intrinsically pro-inflammatory; furthermore, this supports the idea that reshaping of local microbiota and the induction of T$_{regs}$ rescue Nlrp3$^{R258W}$ mice from detrimental inflammation. Further, the Nlrp3$^{R258W}$ x Rag1$^{-/-}$ mice identified in our study represent a promising animal model for colitis caused by innate immune signaling, providing a new option for future studies in intestinal homeostasis and disease.

**Fig. 7** The reshaped microbiota in Nlrp3$^{R258W}$ mice resists gut inflammation through boosting local Tregs. **a** Colonic lamina propria cells were isolated from either cohoused or singly housed WT/Nlrp3$^{R258W}$ mice, intracellular Foxp3 level from CD4$^+$ T cells was analyzed through flow cytometry. The representative samples were shown and the percentage of Foxp3$^+$ cells in CD4$^+$ T cells was summarized. The data are shown as the means ± SEM, *$P <$ 0.05, **$P <$ 0.01, ***$P <$ 0.001 (repeated measures ANOVA with Dunnett's post-hoc test). **b** Feces from the WT and Nlrp3$^{R258W}$ mice were transplanted to germ-free WT mice (Gnoto-+/+, $n = 10$, Gnoto-+/R258W, $n = 11$), colonic lamina propria cells were isolated and intracellular Foxp3 level from CD4$^+$ T cells was analyzed through flow cytometry. The representative samples were shown and the percentage of Foxp3$^+$ cells in CD4$^+$ T cells was summarized. The data are shown as the means ± SEM, *$P <$ 0.05 (two-tailed Student's t test) **c** Colonic lamina propria cells were isolated from WT, Nlrp3$^{R258W}$, Nlrp3$^{R258W}$ x Il1r1$^{-/-}$, and Il1r1$^{-/-}$ mice, intracellular Foxp3 level from CD4$^+$ T cells was analyzed through flow cytometry. The representative samples were shown and the percentage of Foxp3$^+$ cells in CD4$^+$ T cells was summarized. The data are shown as the means ± SEM, *$P <$ 0.05, **$P <$ 0.01 (one-way ANOVA with Tukey's post-hoc test). **d, e** WT untreated mice ($n = 5$), Nlrp3$^{R258W}$ mice i.p. injected with isotype antibody ($n = 4$) and Nlrp3$^{R258W}$ mice i.p. injected with anti-CD25 antibody ($n = 6$) on day −4 and day −2 with 500 μg/mouse/injection were subjected for colitis induction with 4% DSS from day 0. The body weights (**d**) and disease scores (**e**) were assessed daily. Data are shown as the means ± SEM. *$P <$ 0.05, **$P <$ 0.01 (two-tailed Student's t test)

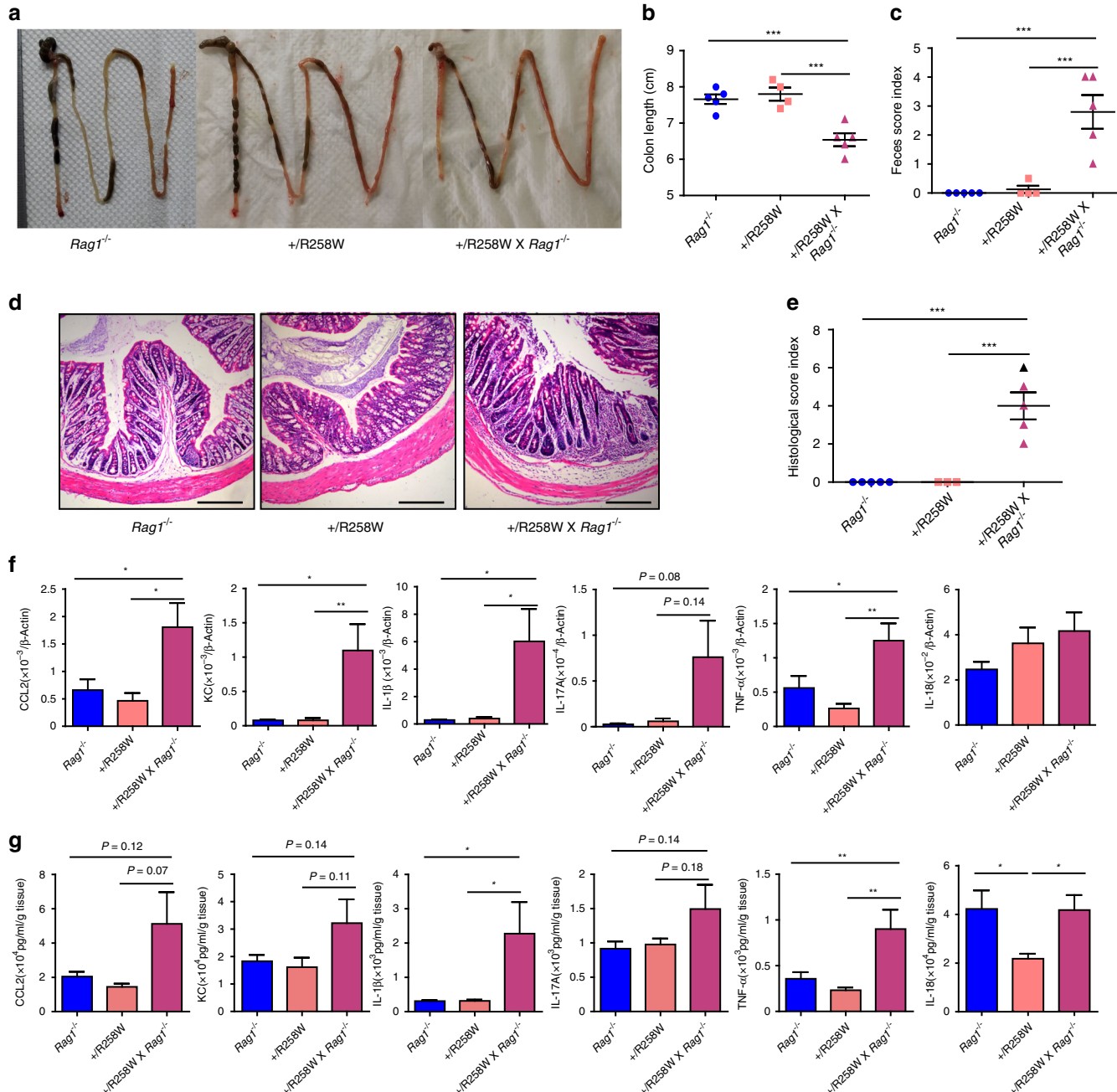

**Fig. 8** The $Nlrp3^{R258W}$ x $Rag1^{-/-}$ mice develop spontaneous colitis. **a–e** Characterization of the spontaneous colitis in $Nlrp3^{R258W}$ x $Rag1^{-/-}$ mice ($n = 5$) together with $Rag1^{-/-}$ ($n = 5$) and $Nlrp3^{R258W}$ ($n = 4$) mice as control, the morphology (**a**), colon length (**b**), feces score (**c**), and histological examination (**d**), scale bar is 200 μm, and histological score (**e**) of $Rag1^{-/-}$, $Nlrp3^{R258W}$, and $Nlrp3^{R258W}$ × $Rag1^{-/-}$ mice were monitored. **f**, **g** The indicated inflammatory cytokine/chemokine expression (**f**) and secretion (**g**) from the colon tissue of indicated mice were determined through Q-PCR and ELISA, respectively. Data are shown as the means ± SEM. *$P < 0.05$, **$P < 0.01$, ***$P < 0.001$ (one-way ANOVA with Dunnett's post-hoc test)

Finally, we are not able to determine in humans whether there are similar beneficial effects endowed by enhanced auto-inflammatory signals as we identified in $Nlrp3^{R258W}$ mice. However, there are reports that despite the auto-inflammatory conditions in skin, joints, and eyes, no gut symptoms were reported in CAPS patients[16–19]. Thus, further validation of this beneficial effect by enhanced auto-inflammatory signals on these or other similar patient cohorts or mouse models is necessary to further the understanding of intestinal homeostasis in these hosts.

In summary, our work uncovers a new form of host–microbiota interaction, in which the genetic defects in the host innate immunity are not necessarily destined to disrupt the gut homeostasis, but may rather improve the original symbiotic microbiota to reach an alternative homeostasis, which can even better defend exogenous inflammatory insults. This finding provides a novel theoretical basis for potential clinical treatments of inflammatory gut diseases through properly manipulating either/both host inflammatory response or/and gut microbiota.

## Methods

**Mice**. All mice used in this study were male mice on C57BL/6 background at ages indicated in specific experiments. $Nlrp3^{R258W}$ (+/R258W) mice and $Nlrp3^{-/-}$ mice

were reported previously[21, 53]. $Nlrp3^{R258W}$ mice were maintained as heterozygotes by backcrossing with WT (+/+) control mice from the Jackson Laboratory. All $Nlrp3^{R258W}$ and WT mice used in this study are littermate controls. $Il1r1^{-/-}$, $Rag1^{-/-}$, and $Casp1/11^{-/-}$ mice were from the Jackson Laboratory. $Nlrp3^{eGFP}$ reporter mouse line was reported before[24]. All mice were maintained in specific pathogen-free facility under a strict 12 h light cycle (on at 0700 hours and off at 1900 hours), and given a regular chow diet ad libitum (#M01-F25, Shanghai SLAC Laboratory Animal Co.). Cohousing (ch-) of WT and $Nlrp3^{R258W}$ mice were carried out at 1:1 ratio since weaning (3–4 weeks) till adulthood (10–12 weeks), or the mice were separated upon weaning according to genotype, denoted as singly housed (sh-). Germ-free mice on C57BL/6 background were purchased from Shanghai SLAC Laboratory Animal Co., and maintained in a germ-free facility. All animal experiments were performed in compliance with the guidelines from the national care and use of laboratory animals and were approved by the institutional ethics committee of the Institut Pasteur of Shanghai, Chinese Academy of Sciences.

**Induction of colitis.** Ten- to 12-week-old male mice (WT ($n = 32$) and $Nlrp3^{R258W}$ ($n = 21$) in Fig. 2a–e; WT ($n = 25$(sh- and ch-)) and $Nlrp3^{R258W}$ ($n = 22$(sh- and ch-)) in Fig. 5a–e; WT ($n = 8$), $Il1r1^{-/-}$ ($n = 9$), $Nlrp3^{R258W}$ X $Il1r1^{-/-}$ ($n = 6$), and $Nlrp3^{R258W}$ ($n = 5$) in Fig. 6e; WT ($n = 8$), $Il18^{-/-}$ ($n = 10$), $Nlrp3^{R258W}$ x $Il18^{-/-}$ ($n = 7$), and $Nlrp3^{R258W}$ ($n = 5$) in Supplementary Fig. 6e, f) were given 2.5% DSS (MW 36 ~ 50 kD, MP Biomedicals) in drinking water at indicated concentration ad libitum for 5 days, followed with regular water for another 2 or 3 days. The DSS solutions were made fresh on day 3. Mice were killed for tissue analyses on day 7 or 8. Then colon tissues were subjected to organ culture, histopathological, and quantitative PCR analyses. For characterization of the spontaneous colitis (Fig. 8) in $Nlrp3^{R258W}$ x $Rag1^{-/-}$ mice ($n = 5$), $Rag1^{-/-}$ ($n = 5$) and $Nlrp3^{R258W}$ ($n = 4$) mice served as control, similar set of analyses were done as in the DSS colitis experiments.

**Induction of colitis-associated colon cancer.** Ten- to 12-week-old male mice (WT ($n = 10$) and $Nlrp3^{R258W}$ ($n = 11$) in Fig. 2i–l; WT ($n = 12$(sh- and ch-)), and $Nlrp3^{R258W}$ ($n = 12$(sh- and ch-)) in Fig. 5f, g) were first i.p. injected with AOM (Sigma) at 7.5 mg/g body weight, 5 days later the mice were given three cycles of 2% DSS in drinking water for 5 days followed by normal drinking water for 2 weeks. The DSS solutions were made fresh on day 3 of each cycle. Body weight change was recorded at indicated time points, final tumor load and tumor size were measured under a microscope.

**Histology.** Colon or small intestine tissues for histological analyses were dissected from indicated mice, immediately fixed in 4% paraformaldehyde. Paraffin-embedded sections of the indicated tissues were subjected to H&E or IHC staining and then examined under a light microscope. To determine the cell proliferation rate, we used Ki67 staining (Cell Signaling Technology, Clone D3B5, 1:1500 dilution). For the statistical analysis of Ki67-positive cells, in total 40 full crypts from 4 animals of each genotype were counted. Histological score for H&E examination was determined as sum of epithelium score and infiltration score.

**RNA isolation and quantitative real-time PCR.** Total RNA was extracted from the indicated cells or mouse colon tissues with TRI reagent (Sigma) according to the manufacturer's instructions. RNA samples were reverse-transcribed into cDNA with a GoScript™ Reverse Transcription kit (Promega). The cDNA samples were amplified by real-time PCR with a SYBR Green kit (Toyobo) on an ABI 7900 HT Fast Real-Time cycler (Applied Biosystems). The expression of target genes was normalized to expression of housekeeping gene beta actin. In Figs. 2h and 6h, arbitrary units (AU) were introduced to normalize the difference between different batches of samples. Primers used were listed in Supplementary Table 3.

**ELISA.** To measure cytokine production, tissues were weighed, washed in sterile phosphate-buffered saline, and then incubated for 24 h at 37 °C with 5% $CO_2$ in complete IMDM supplemented with 10% FBS and penicillin (200 U/ml) + strep-tomycin (200 μg/ml) (Gibco). Mouse cytokines and chemokines in organ culture supernatants were measured with ELISA kits (eBioscience) according to the manufacturer's instructions. Cytokines or chemokines secreted from isolated cells were also determined through ELISA.

**In vivo intestinal permeability assay.** In vivo assay to assess epithelial barrier permeability was performed with a FITC-labeled Dextran (Mw 4000; Sigma-Aldrich) method as described before[48]. Briefly, 10- to 12-week-old male mice (WT ($n = 4$) and $Nlrp3^{R258W}$ ($n = 4$) in Supplementary Fig. 2f; WT ($n = 6$ for 3 days, $n = 3$ for 5 days) and $Nlrp3^{R258W}$ ($n = 3$ for 3 days, $n = 3$ for 5 days) in Fig. 2f) were gavaged with 250 μl 80 mg/ml FITC-Dextran and 4 h later the blood serum was collected upon killing. The leakage of FITC-Dextran into serum was measured with a fluorescence spectrometer at 490 nm excitation and 530 nm emission detection.

**Flow cytometry.** Isolated intestinal lamina propria cells were washed with FACS buffer (1× PBS containing 0.5% BSA and 2 mM EDTA), non-specific binding was blocked with FcR blocking antibody (anti-Mouse CD16/32, eBioscience, dilution

1:100) before staining with labeled monoclonal antibodies against CD45.2 (104, BD Pharmingen, dilution 1:160), MHCII (M5/114.15.2, BioLegend, dilution 1:160), CD11c (N418, eBioscience, dilution 1:160), CD11b (M1/70, BD Pharmingen, dilution 1:160), Ly6G (RB6-8C5, eBioscience, dilution 1:160), Ly6C (AL-21, BD Pharmingen, dilution 1:160), CD3 (145-2C11, BD Pharmingen, dilution 1:160), CD4 (RM4-5, BD Pharmingen, dilution 1:160), and CD64 (X54-5/7.1, BioLegend, dilution 1:160). Intracellular staining was carried out after permeabilization with Permfix (BD) for cytokines IL-17A (eBio17B7, eBioscience, dilution 1:100), IFN-γ (XMG1.2, eBioscience, dilution 1:100); or Transcription Factor Staining Buffer Set (eBioscience) for Foxp3 (FJK-16s, eBioscience, dilution 1:100)/IL-10 (JES5-16E3, eBioscience, dilution 1:100) staining. Dead cells were excluded with a fixable via-bility dye (eBioscience, dilution 1:500) before fixing. The fluorescence was exam-ined on an LSRFortessa™ Flowcytometer (BD).

**Western blot.** LPMCs culture supernatant (serum free) were precipitated with methanol and chloroform for secreted proteins, and the cell lysates were collected by direct lysis by 1x Laemmli loading buffer. Samples were boiled to denature at 100 °C for 10 min before the SDS-PAGE gel electrophoresis for protein separation. After electrophoresis, the protein was wet-transferred to a 0.45 μm nitrocellulose membrane (Millipore), and then blotted for indicated proteins using the antibodies as following: mouse caspase-1 p10 (sc-514, dilution 1:400), mouse IL-1β (sc-7884, dilution 1:400), and ASC (sc-22514, dilution 1:400) were from Santa Cruz, NLRP3 (Cryo2, dilution 1:1000) was from Adipogen, and GAPDH (ap0063, dilution 1:10000) was from Bioworld. The uncropped blot images can be found in Sup-plementary Fig. 8.

**Cell isolation and culture.** BMDM were prepared as described previously[21]. In brief, bone marrow (BM) cells were cultured in Petri dishes in 10 ml of complete IMDM medium in the presence of recombinant M-CSF containing L929-cell-conditioned supernatant. After 3 days of culture, medium was replaced by fresh medium. On day 6, adherent cells were harvested and used as indicated.

To isolate intestinal lamina propria mononuclear cells (LPMCs) and intestinal epithelial cells (ECs), the whole colon (or whole small intestine) was dissected. The superficial adherent adipose tissue was removed with forceps, and then longitude cut open and luminal contents were washed out in sterile PBS. The tissue was cut into 2 cm pieces and incubated in 10 ml per colon/small intestine 1640 RPMI medium containing 5% FBS, 1 mM DTT, and 5 mM EDTA for 20 min at 250 rpm/ 37 °C for three times. The shed cells were taken as epithelial cells. The remaining tissue was washed in PBS and minced into confluent paste, then incubated in 10 ml per colon/small intestine 1640 RPMI medium containing 5% FBS and 2 mg/ml collagenase D (Roche) for 60 min at 250 rpm/37 °C. After digestion, the suspension was centrifuged at 500×g/4 °C for 20 min, and then pellet was resuspended in 3 ml 40% percoll and laid onto 2 ml 70% percoll, 500×g centrifuged for 30 min at minimal acc/dec speed. The interface layer was isolated as LPMCs.

**Fecal bacteria culture.** Stool pellets (~0.05 g/mice) were collected in 1 ml of Ringer's de solution (for 1 l preparation, NaCl 9 g, KCl 0.4 g, $CaCl_2·2H_2O$ 0.25 g, and L-Cystine·HCl 0.5 g, 121 °C autoclaved for 15 min) and homogenized. Dif-ferent dilutions of the suspensions were plated on 5% defibrinated sheep blood trypticase soy agar (Fisher Scientific) and incubated at 37 °C either in an anaerobic jar (Mitsubishi Gas Chemical Co) or normal aerobic condition for 48 h. Bacterial counts were determined by colony-forming assay.

**Depletion of commensal bacteria.** Gut microbiota removal was performed as previously described[54]. Briefly, mice were first treated with a cocktail of antibiotics containing ampicillin (1 g/l), metronidazole (1 g/l), neomycin (1 g/l), and vanco-mycin (0.5 g/l) in drinking water for 4 weeks. The mice continued to receive another cocktail of antibiotics containing streptomycin (2 g/l), ciprofloxacin (0.17 g/l), gentamycin (0.125 g/l), and bacitracin (1 g/l) for 1 week further. All the antibiotics used here were purchased from Sangon Biotech (Shanghai) Co. The depletion efficiency (>99.9%) was determined as described in "Fecal bacteria cul-ture" section.

**Fecal transplantation.** Fresh fecal pellets from singly housed WT control or $Nlrp3^{R258W}$ mice were immediately weighed and placed into Ringer's solution, then diluted to 10 mg/ml, stored at −80 °C before use. Five–six weeks male germ-free mice randomly divided into two groups (Gnoto-+/+ ($n = 4$) and Gnoto-+/R258W ($n = 5$) in Fig. 5h–k; (Gnoto-+/+ ($n = 10$) and Gnoto-+/R258W ($n = 11$) in Fig. 7b) were gavaged (200 μl per mouse) with freshly thawed WT or $Nlrp3^{R258W}$ fecal solution, respectively. Two days later, these mice received another gavage to exclude possibility of any unsuccessful inoculation. After transplantation, mice were continued to be housed in separated isolators for another 3 weeks, during which feces were collected twice a week to monitor the establishment of microbiota colonization.

**In vivo depletion of T$_{regs}$.** Ten-week-old $Nlrp3^{R258W}$ male mice were i.p. admi-nistrated with anti-CD25 antibody (BE0012, BioXCell) ($n = 6$) or isotype control

Ab (BE0088, BioXCell) ($n = 4$) twice at 500 µg/mouse/injection in PBS on day −4 and day −2 (on day 0 start the DSS colitis induction).

**16S rRNA gene V3–V4 region sequencing.** Frozen fecal samples were processed for DNA isolation as previously described[55]. A sequencing library of V3–V4 regions of the 16S rRNA gene was prepared as described in http://res.illumina.com/documents/products/appnotes/16s-metagenomic-library-prep-guide.pdf. Phanta® Super-Fidelity DNA polymerase (P505, Vazyme, China) was used for two steps of amplification. For amplification of 16S rRNA V3–V4 region, the primer 5′-TCGTCGGCAGCGTCAGATGTGTATAAGAGACAGCCTACGGGNGGCWGC AG-3′ and 5′-GTCTCGTGGGCTCGGAGATGTGTATAAGAGACAGGACTAC HVGGGTATCTAATCC-3′ were used with PCR program as starting with pre-denaturation at 94 °C for 3 min, followed by denaturation at 94 °C for 30 s, annealing at 55 °C for 30 s, and extension at 72 °C for 30 s for 20 cycles with a final extension step at 72 °C for 8 min. The AMPure XP beads (Agencount AMPure XP kit, A63881, Beckman) were used to purify the 16S V3–V4 amplicon and the purified amplicon was used for the index PCR (attachment of dual indices and Illumina sequencing adapter using the Nextera XT Index Kit). The purification and index PCR were both carried according to the protocol. The purified products were mixed at equal ratio for sequencing using the Miseq Reagent Kit v3 (600 cycle, MS-1023003, Illumina) on Illumina MiSeq System (Illumina Inc., USA).

**Analysis of 16S rRNA V3–V4 sequencing data.** Both the forward and reverse ends of the same read were truncated at the first base, where the Q value was no more than 2. If the pair of reads had a minimum overlap of 50 bp, they were then merged into a complete read. These reads were not kept unless longer than 399 bp with an expected error of no more than 0.5[56]. Quality-filtered reads were dereplicated into unique sequences and then sorted by decreasing abundance, and singletons were discarded. The representative non-chimeric OTU sequences were next picked by Uparse's default[57]. Further reference-based chimera detection was performed by UCHIME[58] against the Ribosomal Database Project classifier database (v9)[59]. The OTU table was finalized by mapping quality-filtered reads to the remaining OTUs with the Usearch global alignment algorithm at a 97% cutoff. The representative sequences for each OTU were built into a phylogenetic tree by FastTree and assigned the taxonomic classification with the Ribosomal Database Project (RDP Release 11) Classifier with a minimum bootstrap threshold of 80%.

The sequences of all the samples were downsized to 9000 (1000 permutations) to equal the difference in sequencing depth. The alpha diversity of each sample with Faith's phylogenetic diversity (PD_Whole tree), observed OTUs and the Shannon index and the beta diversity based on Bray–Curtis distance were performed on QIIME platform (version 1.91)[60].

RDA was performed using the RDA command of the "vegan" package (2.3-0) of R statistical software (3.2.2) on Hellinger-transformed abundance of OTUs[61]. To evaluate the effect of $Nlrp3^{R258W}$ mutation posed on the gut microbiota, we defined and computed the WT-index for each individual on the basis of the selected key OTUs (explained more than 10% of the variability of the samples) by RDA analysis. For each individual sample, the WT-index of sample $J$ that denoted by $I_j$ was computed by the formula below:

$$I_j\mathrm{wt} = \sum A_{ij}\, i \in N$$

$$I_j\mathrm{ki} = \sum A_{ij}\, i \in M$$

$$I_j = I_j\mathrm{wt}/|N| - I_j\mathrm{ki}/|M|$$

Where $A_{ij}$ is the Hellinger-transformed abundance of OTU $i$ in sample $j$. $N$ is a subset of all wide-type enriched OTUs in these key OTUs. $M$ is a subset of all mutation-enriched OTUs in these key OTUs. And $|N|$ and $|M|$ are the sizes of these two sets. By using the "pROC" package of R software, we then computed the 95% CI of the AUC with 1000 bootstrap replicates to assess the variability of the measure.

Correlation-based networks reflecting co-variations of bacterial abundances were used as a surrogate for bacterial interaction networks. The OTUs shared by more 40% of the samples from each group (single-house +/R258W, single-house +/+, co-house +/R258W, and co-house +/+) during DSS treatment were used to construct interaction networks and the correlation coefficient R for all possible pairs of OTUs were computed by SparCC algorithm. We used 1000 random permutations and set an edge, if $P < 0.05$ and $|R| > 0.7$. The networks were visualized in Cytoscape v3.2.0. Eigenvector centrality was estimated as described and calculated by CytoNCA[62]. Kolmogorov–Smirnov tests were used to test for differences in distributions and performed by R software.

**Metagenomic sequencing.** Ten samples (+/+, $n = 5$ at 10 week of age; and +/R258W $n = 5$ at 10 week of age) were sequenced using the Illumina Hiseq 2000 platform at Shanghai Biotechnology Co., Ltd. DNA library preparation followed Illumina's instructions. Cluster generation, template hybridization, isothermal amplification, linearization, blocking, and denaturing and hybridization of the

sequencing primers were performed according to the workflow indicated by the provider.

Libraries were constructed with an insert size of ~350 bp, followed by high-throughput sequencing to obtain paired-end reads with 150 bp in the forward and reverse directions. An average of $100.6 \pm 30.0$ million (mean ± s.d.) reads were obtained for each sample.

**Analysis of metagenomic sequencing data.** Prinseq[63] was employed (a) to trim the reads from the 3′ end until reaching the first nucleotide with a quality threshold of 20; (b) to remove read pairs if either read was shorter than 60 bp or contained "N" bases; and (c) to de-duplicate the reads. Reads that could be aligned to the host genome (M. musculus, UCSC mm10) were removed (aligned with Bowtie2[64], using ─reorder─no-hd─no-contain─dovetail). On average, $88.4 \pm 4.2$ million (mean ± s. d.) reads for each sample were retained and used for further analysis. Metagenomic data were metabolically profiled by using HUMAnN[65]. All high-quality reads were aligned with Bowtie2 to the KEGG database[66] (Release: 24 February 2014), from which sequences of eukaryotes had been excluded. The alignments were transformed into bam format with SAMtools[67] and input into HUMAnN[65] to obtain the abundance of KOs and pathways. The linear discriminant analysis (LDA) effect size (LEfSe)[28] was used to identify the enzymatic pathways that were significantly different between +/+ and +/R258W mice (threshold on the logarithmic LDA score for discriminative features ≥3; alpha value for the factorial Kruskal–Wallis test between groups <0.05).

**RNAseq and differential expression analysis.** Colon tissues were preserved in RNAlater solution (Ambion) and RNA was extracted by homogenization in Trizol reagent (Sigma) according to the manufacturer's instruction. The extracted RNA was sent for quantification, qualification, library preparation, and sequencing by Novogene Co. on an Illumina Highseq platform and paired-end reads were generated. Raw reads of fastq format were first processed through in-house perl scripts (by Novogene Co.). In this step, clean reads were obtained by removing the reads containing adapter, reads containing ploy-N and low-quality reads. At the same time, Q20, Q30, and GC content of the clean reads were calculated. The clean reads from each sample were aligned to Mus musculus reference genome, GRCm38 (Genome Reference Consortium mouse build 38, ftp://ftp.ensembl.org/pub/release-80/fasta/mus_musculus/dna/), using HISAT (hierarchical indexing for spliced alignment of transcripts, version 0.1.7-beta)[68]. HTSeq-count (vesion 0.6.1p1)[69], an independent application developed in HTSeq python framework, was used to calculate the gene counts of aligned reads through annotation of the mouse genome (ftp://ftp.ensembl.org/pub/release-80/gtf/mus_musculus). After obtaining the gene counts of each sample, statistical comparison between different groups was conducted using DESeq2 (version 1.9.23)[70] with standard parameters. To assess the significance of differential gene expression, the threshold of BH adjusted P value was set at ≤0.05 and the absolute value of $\log_2$-fold change ratio between two groups at ≥0.2. The significant differential genes were annotated by in-house perl scripts. The normalized gene count matrixes generated by DESeq2 during differential expression analysis were abstracted for further analysis.

**Statistics.** The statistic tests were clarified in related text or legends, all tests are two-sided unless otherwise specifically explained.

**Data availability.** The raw Illumina read data for all samples has been deposited in the sequence read archive under the accession number SRP072535. The in-house computer programs used in this study are available from the corresponding author on request with no restrictions. The authors declare that all other data supporting the findings of this study are available within the article and its Supplementary Information files, or are available from the authors on request.

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

## Acknowledgements

This work was supported by grants from National Key Basic Research Programs (2015CB554302 and 2014CB541905), Natural Science Foundation of China (91429307, 31370892, 31570895, 31330005, 81401141 and 81761128012), Strategic Priority Research Program (XDPB0303), and International Partnership Program (153831KYSB20160009) of the Chinese Academy of Sciences, as well as the Science and Technology Commission of Shanghai Municipality (14YF1402200).

## Author contributions

L.Z. and G.M. designed and supervised the study. X.Y. and Y.X. conducted the animal trial, sample collection, and physiological data analysis. X.Y., Y.X., C.Z., Q.Z., F.P. and X.H. prepared the bacterial DNA and conducted the amplicon sequencing. C.Z. and G. W. performed the microbiota data analysis. G.X., Q.Z., Q.G., A.L. and Z.T. helped with animal experiments. B.Z. and H.W. helped with germ-free experiments. X.Z., R.Z. and W.S. provided critical experimental materials. X.Y., C.Z., Y.X., L.Z. and G.M. wrote the manuscript.

## Additional information

**Competing interests:** The authors declare no competing financial interests.

