## [Peer Review File · Nature Communications]

Reviewers' comments:

Reviewer #1 (Remarks to the Author):

Canonical inflammasome activation via IL-18 and IL1B has been implicated in driving colitis in humans. IL-1R blockade has shown therapeutic benefit in case series. Conversely, a gain-of-function mutation in NLRP3 leads to inflammasome activation and systemic, but not gut, autoinflammatory disease. Prior studies utilizing Nlrp3 deficient mice have yielded conflicting results regarding a protective or detrimental role in colitis. This is highly significant, as both specific inflammasome inhibitors and downstream IL-1R blockade are under study for broader treatment of IBD.

The authors have addressed this by examining recovery from DSS colonic injury in mice carrying a gain-of-function mutation in Nlrp3. They show that these mice exhibit an increase in basal IL1b, but not IL-18, production, and a profound shift in basal fecal microbial composition, function, and interconnections. This is associated with protection from colonic injury following acute DSS administration. Several complementary approaches using co-housing, antibiotic administration, and colonization of germ-free mice are used to show that the microbiota of mice carrying mutated Nlrp3 exerts a protective effect, and compound transgenic mice are used to show that this effect upon the microbiota depends upon intact IL-1 signaling. Studies using RNASeq and PCR demonstrate an increased in anti-microbial peptide expression, suggesting a role in modulating the microbiota. Finally, studies utilizing CD25 blockade and quantification of colonic foxp3+ Tregs are used to conclude that microbiota-induced Tregs contribute to the amelioration of the colonic injury.

Prior studies using Nlrp3 deficient mice have implicated alterations in the enteric microbiota, IL-18 enhancement of epithelial wound healing, and IL-10 production by foxp3+ Tregs as potential mechanisms by which Nlrp3 might exert a beneficial effect in the setting of acute colitis. Conversely, with more chronic colitic injury a pro-inflammatory effect of Nlrp3 has been shown, potentially via IL-18 and IL-1b dependent epithelial injury and expansion of IFNG producing effector Th1 lymphocytes.

The current report is the first to my knowledge to test the effect of a gain-of-function mutation in Nlrp3 upon the response to acute colonic injury. It is therefore highly relevant to understanding the surprising lack of gut injury in patients with similar mutations. The main novel aspect of the study is in defining in depth the effects of the Nlrp3:IL1 axis upon the commensal flora, and demonstrating through several complementary approaches its protective properties in the setting of acute colonic injury. These results will be of great interest to the community, given the divergent results of prior studies using Nlrp3 deficient mice, and the fundamental importance of inflammasome biology for mucosal homeostasis and potential novel IBD therapies.

However, there are some concerns that limit support for the authors' conclusions.

1) The data regarding the basal effect of the Nlrp3:IL1 axis upon a protective microbiota are convincing and novel. However, I am not sure that protective effects of Nlrp3:IL-18 following DSS injury have been excluded. It would have been presumed based on prior studies that this was the primary protective effect. It would be quite useful to formally exclude a role for IL-18, at least via blockade in the Nlrp3R258W mice in the setting of DSS administration.

2) The RNASeq experiment could have been quite helpful in discovering the key host:microbe protective mechanisms, but ideally should be performed with a larger number of WT and Nlrp3R258W mice in order to permit a more comprehensive analysis of the colonic global pattern of gene expression to provide better insight into potential protective pathways. Simply show a few antimicrobial peptides is not convincing in terms of establishing a likely Nlrp3:IL1 mechanism for shaping the commensal flora. A more comprehensive gene expression analysis would then also

permit a formal multivariate analysis of covariation of the microbiota and host genes, likely leading to more compelling protective pathways. For example, it appears quite likely that microbiota derived metabolites are exerting a beneficial effect on the wound healing response to the DSS injury, and studies to directly test that would have been of great interest.

3) The final data set regarding a potential protective role of microbiota induced foxp3+Tregs is not convincing. This would need to at least be repeated with a greater sample size to be sure that it is reproducible, with appropriate controls for the effect of the CD25 blockade. It would be surprising if the predominant protective mechanism in the setting of acute DSS injury involved Tregs. Alternately, this potential mechanism could be left for another study which included a more clearly effector T cell dependent model, such as T cell transfer colitis.

4) The protective versus detrimental effect of Nlrp3 in the setting of DSS colitis has been remarkably sensitive to experimental protocols, but overall seems to favor a protective role with more acute injury as in the current study, and a detrimental role with more chronic injury. In that sense, it would be quite helpful to expose the Nlrp3R258W mice to a more chronic DSS protocol (as reviewed in Pellegrini et al, Canonical and Non-Canonical Activation of NLRP3 Inflammasome at the Crossroad between Immune Tolerance and Intestinal Inflammation", Table 1) to determine whether this leads to a shift to exacerbation of inflammation, as reported for Nlrp3 deficient mice, or whether the microbiota continues to exert a protective effect even under conditions of more chronic injury.

5) The microbial networks are quite interesting, but should please be provided in a larger font which is easier to read.

6) The statistical analyses need to be clarified, as it appears that t-tests have been used when repeated-measures ANOVA, or simple ANOVA, would have been more appropriate for comparisons involving multiple groups. It does not appear that a correction for multiple comparisons has been applied when appropriate, and it would be surprising if most of the data were normally distributed as suggested.

7. It would be helpful to show IL18 in Figure 1.

8. Why were female mice not studied?

9. A supplemental Table for the Fig. 3c OTUs would be helpful.

10. Please clarify the mouse diet used.

11. Figure 4f typo: blue lines "negative" correlations

Reviewer #2 (Remarks to the Author):

General comments:

The authors studies gut physiology in a mouse model bearing a point mutation with gain of function in NLRP3 which is a critical component of the NALP3 inflammasome. In the mouse they found signs of skin inflammation as in cryopyrin-associated periodic syndromes but no signs of colitis. In contrast, the authors describe a "stronger resistance to colitis and CRC" due to increased expression of antibacterial peptides and changes in the microbiota. The manuscript is interesting (despite the lack of a phenotype) and the experiments are well performed (cage controls and littermate controls). Some data need to be added to make the manuscript completely convincing.

Specific comments:

- 1) The authors describe that in their model they only found NLRP3 expression in lamina propria cells. They show data on isolated cells. Some additional data would be helpful to support their finding. There are good reports indicating an important role of the NLRP3 inflammasome in intestinal epithelial cells (e.g. George X. Song-Zhao, et al., Kevin J. Maloy, Irf3 Activation in the Intestinal Epithelium Protects against a Mucosal Pathogen; *Mucosal Immunol.* 2014 Jul; 7(4): 763–774.) In fact, this group already has shown that NLRP3 activation protects from colitis. Some morphological data on the presence of NLRP3 expression would be helpful. Is it possible that the point mutation changes the expression pattern? Why would expression in IEC be absent? What was the change of IL-18 expression under inflammatory conditions?
- 2) As IL-18 has been shown to play an important role in mucosal defense mechanisms, the addition of data on IL-18 secretion would be important. As intestinal epithelial cells (IEC) are an important source of IL-18 and IL-18 is also processed by the NLRP3 inflammasome those data would also add to the question mentioned in point 1.
- 3) NLRP3 inflammasome and autophagy functions are closely connected. How was autophagy changes in the cells with over-activation of NLRP3?
- 4) It is impossible to say that the colon of the *Nlrp3*R258W mice was “free of pathologies”. With the methods used the authors did not find any alterations. However, there could be differences in innate immune functions that don’t lead to any morphological changes. The statement needs to be toned down. What about IL-18 levels in the tissue?
- 5) The authors only performed acute DSS colitis. A model for chronic intestinal inflammation would be chronic DSS colitis or spontaneous in *Nlrp3*R258W/*Il10* deficient mice.
- 6) A language editing would certainly be helpful.

Reviewer #3 (Remarks to the Author):

In humans, dysregulation of inflammasome signalling by gain-of-function mutations in the coding region of NLRP3 results in a group of diseases called cryopyrin-associated periodic syndromes (CAPS), which are characterized by auto-inflammatory conditions in skin, joints and eyes. Intriguingly, severity of skin disease was linked to an exacerbated response of mast cells to the skin microbiota in the mice carrying a CAPS-causing mutation, while no gut symptoms were reported. The investigation of the underlying mechanism leading to either disease protection in the gut or susceptibility in the skin in response to a disease-causing *Nlrp3* mutation is of great importance. Collectively, the methodology used in this study is adequate and the conclusions reached in this article are well supported by experimental data. The authors provided experimental evidence that the absence of intestinal inflammation in the mutant mice was a consequence of a greater abundance of regulatory T cells in response to several changes in the composition of its gut microbiota. Mechanistically, the authors demonstrated that exacerbated interleukin-1 signaling activation in mice carrying the *Nlrp3* R258W mutation results in downstream activation of several antimicrobial peptides and in subsequent changes in the composition of the gut microbiota that protects mice against intestinal inflammation. Here, there are some concerns that, I think, need to be considered:

- The authors showed a greater ability of colonic CD11b⁺/CD11c⁺ phagocytes to secrete IL1b in response to heat-inactivated fecal bacteria, but it is not yet clear if the latter responsiveness of phagocytes is acquired in response to age-induced changes in the composition of the gut microbiota (eg. after weaning). We may indeed expect that the phenotype is constitutive as mutant mice develop skin disease in response to some microbes when being neonates. Foremost, it would strengthen the manuscript to determine whether intestinal mast cells may secrete greater amount of IL1b in the colon of mutant mice as it was previously shown in the skin (*Immunity.* 2012 Jul 27;37(1):85-95).
- While the authors provide experimental evidence that IL1 signaling is required for protection against colitis by shaping the composition of the gut microbiota in mutant mice, additional experiments are required to determine the role of upstream since neutralization of TNF- α

abrogated IL-1 β production and skin disease in neonatal Nlrp3 mutant mice (Immunity. 2012 Jul 27;37(1):85-95).

- While the authors provided convincing evidence that the Nlrp3 R258W mutation gives a growth advantage to both Clostridia XIVA and Lactobacillus murinus that were found to promote Tregs differentiation, the use of electronic microscopy and of fluorescent in situ hybridization with specific probes targeting either Clostridia or Lactobacilli species would help to determine at which age mutant mice showed a greater colonization with the aforementioned bacteria and where those are colonizing the intestinal mucosa for promoting Tregs induction.

Reviewers' comments:

Reviewer #1 (Remarks to the Author):

Canonical inflammasome activation via IL-18 and IL1B has been implicated in driving colitis in humans. IL-1R blockade has shown therapeutic benefit in case series. Conversely, a gain-of-function mutation in NLRP3 leads to inflammasome activation and systemic, but not gut, autoinflammatory disease. Prior studies utilizing Nlrp3 deficient mice have yielded conflicting results regarding a protective or detrimental role in colitis. This is highly significant, as both specific inflammasome inhibitors and downstream IL-1R blockade are under study for broader treatment of IBD.

The authors have addressed this by examining recovery from DSS colonic injury in mice carrying a gain-of-function mutation in Nlrp3. They show that these mice exhibit an increase in basal IL1b, but not IL-18, production, and a profound shift in basal fecal microbial composition, function, and interconnections. This is associated with protection from colonic injury following acute DSS administration. Several complementary approaches using co-housing, antibiotic administration, and colonization of germ-free mice are used to show that the microbiota of mice carrying mutated Nlrp3 exerts a protective effect, and compound transgenic mice are used to show that this effect upon the microbiota depends upon intact IL-1 signaling. Studies using RNASeq and PCR demonstrate an increased in anti-microbial peptide expression, suggesting a role in modulating the microbiota. Finally, studies utilizing CD25 blockade and quantification of colonic fox3+ Tregs are used to conclude that

microbiota-induced Tregs contribute to the amelioration of the colonic injury.

Prior studies using Nlrp3 deficient mice have implicated alterations in the enteric microbiota, IL-18 enhancement of epithelial wound healing, and IL-10 production by foxp3+ Tregs as potential mechanisms by which Nlrp3 might exert a beneficial effect in the setting of acute colitis. Conversely, with more chronic colitic injury a pro-inflammatory effect of Nlrp3 has been shown, potentially via IL-18 and IL-1b dependent epithelial injury and expansion of IFNG producing effector Th1 lymphocytes.

The current report is the first to my knowledge to test the effect of a gain-of-function mutation in Nlrp3 upon the response to acute colonic injury. It is therefore highly relevant to understanding the surprising lack of gut injury in patients with similar mutations. The main novel aspect of the study is in defining in depth the effects of the Nlrp3:IL1 axis upon the commensal flora, and demonstrating through several complementary approaches its protective properties in the setting of acute colonic injury. These results will be of great interest to the community, given the divergent results of prior studies using Nlrp3 deficient mice, and the fundamental importance of inflammasome biology for mucosal homeostasis and potential novel IBD therapies.

However, there are some concerns that limit support for the authors' conclusions.

- 1) The data regarding the basal effect of the Nlrp3:IL1 axis upon a protective microbiota

are convincing and novel. However, I am not sure that protective effects of Nlrp3:IL-18 following DSS injury have been excluded. It would have been presumed based on prior studies that this was the primary protective effect. It would be quite useful to formally exclude a role for IL-18, at least via blockade in the Nlrp3R258W mice in the setting of DSS administration.

Reply: Thanks for the comment and suggestion. This is an important issue, as the role of Nlrp3:IL-18 axis in experimental DSS colitis and colon cancer had been reported to be protective (Zaki et al, 2010, *Immunity*, doi: 10.1016/j.immuni.2010.03.003; Zaki et al, 2010, *J Immunol*, doi: 10.4049/jimmunol.1002046). As a matter of fact, we have considered this point and crossed the *Nlrp3*^{R258W} mice to *Il18*^{-/-} background followed with DSS colitis experiment. We found that in the absence of IL-18, the *Nlrp3*^{R258W} mutation did not show any protective effect (Supplementary Figs. 6e-6f in our manuscript). So, there is no doubt that IL-18 is protective against colitis development in mice. Interestingly, however, in addition to a basal effect of the Nlrp3:IL-1 β axis upon a protective microbiota under steady state, our further results showed that the production of IL-18 from the gut of experimental mice was dependent on caspase-1, BUT NOT NLRP3 (Fig. 6d, text lines 315-332). Upon DSS treatment for 4 days, IL-18 production was significantly elicited in the gut epithelium, which was diminished in the absence of caspase-1 BUT NOT NLRP3; while in the lamina propria mononuclear cells, IL-18 remained at basal level (Fig. 6d, lower panel). These results suggest that during DSS injury, IL-18 from the epithelium responded quickly, which potentially mediated its protective effect. But the IL-18 production from epithelial cells must have been mediated by some inflammasomes expressed in the epithelium (such as NLRC4/NLRP6/NLRP9b), BUT NOT NLRP3, because NLRP3 is expressed in the lamina propria mononuclear cells, not in gut epithelial cells (Figs. 1a, 6a-6b, supplementary Figs 6a). Of note, during the revision of our current manuscript, using fractionated intestinal cells, a study showed very similar results with our data that NLRP3 is expressed selectively in the lamina propria, while IL-18 is expressed only in the epithelium both in mice and humans (Fig. 2a, Extended Figs. 3a, 3c and 10c from Zhu et al, 2017, *Nature*, doi: 10.1038/nature22967). The reason why *Nlrp3*^{R258W} x *Il18*^{-/-} mice lost resistance to DSS challenge was probably due to a more predominant and frontier role for IL-18 in epithelium than the NLRP3-R258W protein in the lamina propria, because in principle DSS colitis model is to injure the epithelium first, then allow commensal bacteria to activate immune cells in the lamina propria. In summary, our data don't exclude IL-18's essential role in protecting host against DSS colitis, but suggest that the effect of IL-18 is not a direct consequence of NLRP3 inflammasome activation in the intestine.

2) The RNASeq experiment could have been quite helpful in discovering the key host:microbe protective mechanisms, but ideally should be performed with a larger number of WT and Nlrp3R258W mice in order to permit a more comprehensive analysis of the colonic global pattern of gene expression to provide better insight into potential protective pathways. Simply show a few antimicrobial peptides is not convincing in terms of establishing a likely Nlrp3:IL1 mechanism for shaping the commensal flora. A more comprehensive gene expression analysis would then also permit a formal multivariate analysis of covariation of the microbiota and host genes, likely leading to more compelling protective pathways. For example, it appears quite likely that microbiota derived metabolites are exerting a beneficial effect on the wound healing response to the DSS injury, and studies to directly test that would have been of great interest.

Reply: Thanks for the comments. Our current study is focused on addressing the unique novel working model for the NLRP3 inflammasome in the intestine (e.g. selective regulation of IL-1 β but not IL-18); and on revealing an unprecedented microbiota-host interaction relationship, wherein the flexible microbiota actively regulates host immunity (enhancing Tregs) to compensate otherwise detrimental inflammation caused by host genetic defect (NLRP3^{R258W}). We appreciate very much of the suggestion that using RNA profiling in combination with microbiota analysis to dissect the correlated pathways as potentially more compelling mechanisms downstream of NLRP3: IL-1 β for reshaping microbiota. Moreover, the speculation that certain microbiota derived metabolites are mediators of inflammation resistance is quite reasonable. In fact, our preliminary result from a direct analysis of fecal and urine metabolites by NMR have shown notable difference between WT and mutant mice derived samples, and several promising metabolites have been revealed for further functional validation (see Fig. R1 on next page please). However, to reach a conclusive theory, both the aforementioned assumptions require a large body of *in vivo* and *in vitro* experiments, which is impossible to accomplish in the revision stage of our present work. So in our humble opinion, also considering the focus of the current study, the data showing that the antimicrobial peptides are the effectors downstream of NLRP3:IL-1 β to reshape microbiota are substantial. We think it is more practical to include further in-depth mechanistic study based on our preliminary data to address the relationship between metabolites and host gene regulations in a separate study in the future.

Figure R1. The metabolomic analysis of feces and urine from WT and *Nlrp3*^{R258W} mice. Feces (a) and urine (b) from WT and *Nlrp3*^{R258W} mice were analyzed with NMR, left panels show the OPLS-DA plot with R2X and Q2 value, P value was from CV-ANOVA test; right panels show the peaks of different metabolites between the two lines of mice, red peaks indicate metabolites that are significantly different (Two tailed t-test, $P < 0.05$).

3) The final data set regarding a potential protective role of microbiota induced foxp3+Tregs is not convincing. This would need to at least be repeated with a greater sample size to be sure that it is reproducible, with appropriate controls for the effect of the CD25 blockade. It would be surprising if the predominant protective mechanism in the setting of acute DSS injury involved Tregs. Alternately, this potential mechanism could be left for another study which included a more clearly effector T cell dependent model, such

as T cell transfer colitis.

Reply: The protective role of Tregs in the acute DSS colitis model had been reported previously (Atarashi et al 2011, *Science*, doi: 10.1126/science.1198469; Wang et al, 2015, *Mucosal Immunology*, doi:10.1038/mi.2015.20). Moreover, adoptive transfer of CD4⁺CD25⁺ Tregs had been shown to inhibit innate inflammatory responses in mice on *Rag* deficient genetic background (Maloy et al, 2003, *J Exp Med*, doi: 10.1084/jem.20021345). Nonetheless, we do agree that the acute DSS colitis model alone may not be enough to reflect the microbiota induced Tregs' anti-inflammatory effect. So we also considered the T cell transfer colitis model as suggested. Actually we had started the crossing of the *Nlrp3*^{R258W} mice to *Rag1*^{-/-} background about 2 years ago. Quite unexpectedly, but also reasonably, after crossing to *Rag1*^{-/-} background, the *Nlrp3*^{R258W} mice developed notable spontaneous colitis (Fig. 8, text lines 395-415). This is a strong evidence to support: (1) Tregs are indeed critical for the restriction of gut inflammation in the *Nlrp3*^{R258W} mice; (2) the *Nlrp3*^{R258W} mutation in the lamina propria of intestine is intrinsically pro-inflammatory, it is thanks to the local microbiota and Tregs that the alternative homeostasis gets established. In addition, anti-CD25 administration in our experiment was efficient to deplete Tregs in the lamina propria, which can be seen in Supplementary Fig. 7a.

4) The protective versus detrimental effect of Nlrp3 in the setting of DSS colitis has been remarkably sensitive to experimental protocols, but overall seems to favor a protective role with more acute injury as in the current study, and a detrimental role with more chronic injury. In that sense, it would be quite helpful to expose the Nlrp3R258W mice to a more chronic DSS protocol (as reviewed in Pellegrini et al, Canonical and Non-Canonical Activation of NLRP3 Inflammasome at the Crossroad between Immune Tolerance and Intestinal Inflammation", Table 1) to determine whether this leads to a shift to exacerbation of inflammation, as reported for Nlrp3 deficient mice, or whether the microbiota continues to exert a protective effect even under conditions of more chronic injury.

Reply: Thanks for the suggestion. Actually, the AOM+DSS induced colorectal cancer model, with lower amount (2%) and repeated 3 cycles of DSS administration, can be taken as a

good reflection of chronic colitis model (Allen et al, 2010, J Exp Med, doi: 10.1084/jem.20100050). In our experiment, the colitis severity developed in the first two cycles was significantly higher in the WT mice, as reflected by the weight loss curve and the early deaths of WT mice in the second cycle of DSS treatment (Fig. 2k-l). These results indicated a similar protective role of *Nlrp3*^{R258W} mutation in the chronic inflammation.

5) The microbial networks are quite interesting, but should please be provided in a larger font which is easier to read.

Reply: We have reformatted the figure as suggested.

6) The statistical analyses need to be clarified, as it appears that t-tests have been used when repeated-measures ANOVA, or simple ANOVA, would have been more appropriate for comparisons involving multiple groups. It does not appear that a correction for multiple comparisons has been applied when appropriate, and it would be surprising if most of the data were normally distributed as suggested.

Reply: Thanks for pointing out the mistake. After careful re-examination, we have done the statistical analysis more properly, details can be found in the relevant text and figure legend.

7. It would be helpful to show IL18 in Figure 1.

Reply: We have added IL-18 secretion in Fig 1b as suggested.

8. Why were female mice not studied?

Reply: The female mice in our hands tend to have mild disease in the DSS colitis model, thus

from the very beginning we kept using male mice to make the data consistent. Because gender difference also have potential to interfere the disease phenotype, as well as the microbiota constitution, it's more reasonable to exclude female mice.

9. A supplemental Table for the Fig. 3c OTUs would be helpful.

Reply: Supplemental table 2 has been added as suggested.

10. Please clarify the mouse diet used.

Reply: The diet clarification has been added in the methods section as suggested.

11. Figure 4f typo: blue lines "negative" correlations

Reply: This has been corrected, thanks.

Reviewer #2 (Remarks to the Author):

General comments:

The authors studies gut physiology in a mouse model bearing a point mutation with gain of function in NLRP3 which is a critical component of the NALP3 inflammasome. In the mouse they found signs of skin inflammation as in cryopyrin-associated periodic syndromes but no signs of colitis. In contrast, the authors describe a "stronger resistance to colitis and CRC" due to increased expression of antibacterial peptides and changes in the microbiota. The manuscript is interesting (despite the lack of a phenotype) and the

experiments are well performed (cage controls and littermate controls). Some data need to be added to make the manuscript completely convincing.

Specific comments:

1) The authors describe that in their model they only found NLRP3 expression in lamina propria cells. The show data on isolated cells. Some additional data would be helpful to support their finding. There are good reports indicating an important role of the NLRP3 inflammasome in intestinal epithelial cells (e.g. George X. Song-Zhao, et al., Kevin J. Maloy, *Irp3 Activation in the Intestinal Epithelium Protects against a Mucosal Pathogen; Mucosal Immunol.* 2014 Jul; 7(4): 763–774.) In fact, this group already has shown that NLRP3 activation protects from colitis. Some morphological data on the presence of NLRP3 expression would be helpful. Is it possible that the point mutation changes the expression pattern? Why would expression in IEC be absent? What was the change of IL-18 expression under inflammatory conditions?

Reply: These are all important questions. Indeed, there are a few studies showed that NLRP3 acted in the nonhematopoietic compartment to protect mice against colitis (Zaki et al, 2010, *Immunity*, doi: 10.1016/j.immuni.2010.03.003; Song-Zhao et al, 2014, *Mucosal Immunology*, doi: 10.1038/mi.2013.94). However, final conclusions from both of these studies were based on bone marrow transplantation (BMT) strategy only, which compromised the reliability of their conclusions. Actually, there is a separate study also using BMT in DSS colitis model, but claimed a major role for the NLRP3 inflammasome in the hematopoietic compartment (Allen et al, 2010, *J Exp Med*, doi: 10.1084/jem.20100050), which was totally opposite to the conclusion from the first two studies. Moreover, the first two studies didn't show any expression evidence for NLRP3 protein in the epithelium directly, despite that a clear ASC expression in the epithelium was shown (Song-Zhao et al, 2014, *Mucosal Immunology*, doi: 10.1038/mi.2013.94). Intriguingly, although they proposed a major role for NLRP3 in the epithelium, Song-Zhao et al found that the epithelium inflammasome activation required ASC but not NLRP3 (Fig. 5g in Song-Zhao et al, 2014, *Mucosal Immunology*, doi: 10.1038/mi.2013.94). Thus, the contradictory results in these

aforementioned studies and the intriguing observation that NLRP3 protein was dispensable for gut epithelium inflammasome activation highlight the current ill-characterization of where and how NLRP3 expresses and functions in the intestine.

Our data using the *Nlrp3*^{eGFP} reporter mice showed clear expression of NLRP3 in the CD11b⁺/CD11c⁺ lamina propria mononuclear phagocytes but not in the epithelium (Fig. 1a, Supplementary Fig. 1a). Our western blot also successfully detected NLRP3 protein in the lamina propria mononuclear cells from both *Nlrp3*^{R258W} and WT mice (Fig. 1c). Moreover, the Q-PCR analysis confirmed such expression pattern of NLRP3, and the *Nlrp3*^{R258W} mutation did not change the expression pattern (Figs. 6a-6b, Supplementary Fig. 6a). As for the suggestion that to provide some morphological evidence of NLRP3 expression in the gut, we tried to detect the eGFP fluorescence signal in situ from the colon cryosections of *Nlrp3*^{eGFP} reporter mice, and indeed found some dot-like signals in the junction-area between muscularis mucosae and lamina propria compartments (arrow heads in Fig. R2, see below), while the epithelium showed none of such signal. As a positive control, we found very bright and ubiquitous signal from a CMV-eGFP reporter mice colon. The results were showed in Fig. R2. Since our FACS, western and QPCR data are consistent and convincing, and we think the in situ signal of the cryosection is too weak to reach publication quality (probably due to the small number of LPMCs under homeostasis), so we decided to show the data only in this response letter.

Figure R2. In situ examination of NLRP3 expression in the mouse colon. CMV-eGFP (positive control), WT (negative control) and *Nlrp3*^{eGFP} mice colons were dissected and cryosectioned, the eGFP fluorescence was monitored, and the nucleus was visualized by DAPI with confocal microscopy. Scale bars, 50um. Arrow heads are indicating the positive signal for NLRP3 in the lamina propria.

The reason why NLRP3's expression is absent in the IEC is currently unknown. But during the revision of the current manuscript, using fractionated intestinal cells, another paper showed very similar results with our data that both mouse and human NLRP3 are expressed selectively in the lamina propria, not in epithelium; while IL-18 is expressed only in the gut epithelium (Figs. 1a, 6a-6b, Supplementary Fig. 6a in our manuscript; and Fig. 2a, Extended Figs. 3a, 3c and 10c from Zhu et al, 2017, Nature, doi: 10.1038/nature22967). Certainly such expression pattern of NLRP3 in the intestine is interesting, while the mechanisms behind require future study, which can be an independent project: expression regulation of NLRP3 in different populations of cells in the gut and related mechanisms.

The change of IL-18 and IL-1 β under inflammatory conditions also intrigued us, so we tested their secretion in mice after 4-day DSS treatment. We found that the cell compartment pattern in which they express didn't change under such inflammatory conditions, however, we did find an increase of both IL-18 in epithelium and IL-1 β in the lamina propria triggered by the DSS challenge, and only the secretion of IL-1 β depended on the NLRP3 inflammasome (Figs. 6c-6d, text lines 315-332).

2) As IL-18 has been shown to play an important role in mucosal defense mechanisms, the addition of data on IL-18 secretion would be important. As intestinal epithelial cells (IEC) are an important source of IL-18 and IL-18 is also processed by the NLRP3 inflammasome those data would also add to the question mentioned in point 1.

Reply: Thanks for the suggestion. Accordingly, in Fig. 1b, we have added the secretion of IL-18, showing its lack of regulation by *Nlrp3*^{R258W} mutation in the lamina propria. Our data in Figs. 6a-6d and in Supplementary Fig. 6b all supported the concept that IEC is the major source for IL-18, and IL-18 is not regulated by NLRP3, because NLRP3 is not expressed in IEC. These results are consistent with previous data, for example, Song-Zhao et al showed that both IL-18 secretion and inflammasome activation in IEC were independent from NLRP3 (Fig. 4a and Fig. 5g from Song-Zhao et al, 2014, Mucosal Immunology, doi: 10.1038/mi.2013.94). In addition, very similar with our data, Zhu et al showed that both mouse and human NLRP3 are expressed selectively in the lamina propria, while IL-18 is expressed only in the epithelium in the gut (Fig. 2a, Extended Figs. 3a, 3c and 10c from Zhu et al, 2017, Nature, doi: 10.1038/nature22967).

3) NLRP3 inflammasome and autophagy functions are closely connected. How was autophagy changes in the cells with over-activation of NLRP3?

Reply: This is an interesting question. As shown in the following Fig. R3, we found a dampened autophagy response in the peritoneal macrophages from our *Nlrp3*^{R258W} mice upon Rapamycin or LPS stimulation, which is reflected by the decreased LC3-II in the *Nlrp3*^{R258W} mutant cells than that in WT cells.

Figure R3. Autophagy induction in WT and *Nlrp3*^{R258W} peritoneal macrophages. The peritoneal macrophages were induced with i.p. injection of 2ml 3% thioglycolate broth per mouse for 4 days, the isolated cells were then stimulated with 1 μ M Rapamycin or 500ng/ml lipopolysaccharide (LPS) for 7hr in the absence or presence of 0.1 μ M Bafilomycin A, which was added 30min prior the stimulation. The cell lysates were collected for western blot detection of autophagy associated protein LC3-I/II. GAPDH was used as loading control.

4) It is impossible to say that the colon of the *Nlrp3*^{R258W} mice was “free of pathologies”.

With the methods used the authors did not find any alterations. However, there could be differences in innate immune functions that don't lead to any morphological changes. The statement needs to be toned down. What about IL-18 levels in the tissue?

Reply: Thanks for the comment. Accordingly, we have changed the term “free of pathologies” to a more moderate one “maintained homeostasis”. The tissue level of IL-18 under resting conditions has also been added in Supplementary Fig. 2g.

5) The authors only performed acute DSS colitis. A model for chronic intestinal inflammation would be chronic DSS colitis or spontaneous in *Nlrp3*^{R258W}/*Il10* deficient mice.

Reply: Thanks for the comment. Actually we have included some chronic intestinal inflammation model in our study: First, in the AOM+DSS induced colorectal cancer model, we treated mice with lower amount (2%) and repeated (3 cycles) administration of DSS, which can be taken as a chronic colitis model (Allen et al, 2010, J Exp Med, doi: 10.1084/jem.20100050). In this model, the colitis severity developed in the first two cycles was significantly higher in the WT mice, as reflected by the weight loss curve and the early deaths of WT mice in the second cycle of DSS treatment (Figs. 2k-2l). These results indicated a similar protective role of *Nlrp3*^{R258W} mutation in the chronic inflammation. Secondly, we also considered the T cell transfer colitis model as a more chronic model to explore whether without Tregs the *Nlrp3*^{R258W} mice would develop stronger disease in this model. Actually, we had started the crossing of the *Nlrp3*^{R258W} mice to *Rag1*^{-/-} background about 2 years ago. Quite unexpectedly, but also reasonably, after crossing to *Rag1*^{-/-} background, the *Nlrp3*^{R258W} mice developed notable spontaneous colitis (Fig. 8, text lines 395-415). This is a strong evidence to support: (1) Tregs are indeed critical for the restriction of gut inflammation in the *Nlrp3*^{R258W} mice; (2) the *Nlrp3*^{R258W} mutation in the lamina propria of intestine is intrinsically pro-inflammatory, it is thanks to the local microbiota and Tregs that the alternative homeostasis gets established. In addition, anti-CD25 administration in our experiment was efficient to deplete Tregs in the lamina propria, which can be seen in Supplementary Fig. 7a.

Together, by using these two different chronic colitis models, we are able to support our conclusion that *Nlrp3*^{R258W} mice maintain homeostasis and resist exogenous inflammatory insults through the reshaped microbiota-induced Tregs besides the same conclusions obtained with acute DSS colitis model.

6) A language editing would certainly be helpful.

Reply: Thanks for the suggestion. We have edited our manuscript with MSC-Scientific Editing service.

Reviewer #3 (Remarks to the Author):

In humans, dysregulation of inflammasome signalling by gain-of-function mutations in the coding region of NLRP3 results in a group of diseases called cryopyrin-associated periodic syndromes (CAPS), which are characterized by auto-inflammatory conditions in skin, joints and eyes. Intriguingly, severity of skin disease was linked to an exacerbated response of mast cells to the skin microbiota in the mice carrying a CAPS-causing mutation, while no gut symptoms were reported. The investigation of the underlying mechanism leading to either disease protection in the gut or susceptibility in the skin in response to a disease-causing Nlrp3 mutation is of great important. Collectively, the methodology used in this study is adequate and the conclusions reached in this article are well supported by experimental data. The authors provided experimental evidence that the absence of intestinal inflammation in the mutant mice was a consequence of a greater abundance of regulatory T cells in response to several changes in the composition of its gut microbiota. Mechanistically, the authors demonstrated that exacerbated interleukin-1 signaling activation in mice carrying the Nlrp3 R258W mutation results in downstream activation of several antimicrobial peptides and in subsequent changes in the composition of the gut microbiota that protects mice against intestinal inflammation. Here, there are some concerns that, I think, need to be considered:

- The authors showed a greater ability of colonic CD11b+/CD11c+ phagocytes to secrete IL1b in response to heat-inactivated fecal bacteria, but it is not yet clear if the latter responsiveness of phagocytes is acquired in response to age-induced changes in the composition of the gut microbiota (eg. after weening). We may indeed expect that the phenotype is constitutive as mutant mice develop skin disease in response to some

microbes when being neonates. Foremost, it would strengthen the manuscript to determine whether intestinal mast cells may secrete greater amount of IL1b in the colon of mutant mice as it was previously shown in the skin (Immunity. 2012 Jul 27;37(1):85-95).

Reply: Thanks for the comment and suggestion. For the question about whether the NLRP3 inflammasome hyperactivity of R258W mutant colonic CD11b⁺/CD11c⁺ phagocytes was a consequence of age-induced changes in the gut microbiota, we used 3-week-old R258W mutant and littermate WT mice to isolate their colonic lamina propria mononuclear cells, and we found that the responsiveness of these young mice derived cells to bacteria stimulation was already robust (Fig. R4a-4b on next page). This result supports the stated expectation and previous observations that neonatal CAPS mice develop skin symptoms already.

To evaluate the role of intestinal mast cells, we examined the NLRP3 expression in these cells. However, as shown in the following Figure R4c, we failed to detect a positive NLRP3 expression signal in these intestinal mast cells, whereas the signal in the lineage⁺ lamina propria cells was clear. This result suggested a limited, if any, role of NLRP3 inflammasome in the intestinal mast cells under homeostasis to shape gut microbiota.

Figure R4. NLRP3^{R258W} Inflammasome activity in intestinal lamina propria mononuclear cells from young mice upon weaning and in intestinal mast cells. (a-b), the intestinal lamina propria mononuclear cells were isolated from just weaned mice (3 weeks old) and stimulated

with heat-inactivated fecal bacteria (BAC, MOI=1: 20) with or without ATP(5mM) pulse, the secreted IL-1 β and TNF- α were determined through ELISA. (c) isolated colonic lamina propria mononuclear cells from mast cell deficient *Kit*^{W-sh/W-sh}, WT and *Nlrp3*^{eGFP} mice were stained with markers for mast cells (gated as CD45.2⁺lin⁻CD117⁺Fcer1 α ⁺ cells). On gated mast cells, the NLRP3 reporter eGFP signal was monitored.

- While the authors provide experimental evidence that IL1 signaling is required for protection against colitis by shaping the composition of the gut microbiota in mutant mice, additional experiments are required to determine the role of upstream since neutralization of TNF- α abrogated IL-1 β production and skin disease in neonatal *Nlrp3* mutant mice (Immunity. 2012 Jul 27;37(1):85-95).

Reply: To test whether TNF- α acts upstream of NLRP3 inflammasome in the lamina propria mononuclear cells, we directly applied TNF- α to stimulate these cells followed with ATP pulse. However, as shown in the following Figure R5, we didn't observe any activation of the NLRP3 inflammasome as indicated by IL-1 β secretion (Fig. R5a on next page). Examination of IL-6, IL-1 β and NLRP3 expression found minimal upregulation of these genes by TNF- α either (Figs. R5b-5d on next page). We validated that the recombinant TNF- α we used was functional, because it successfully stimulated the phosphorylation/activation of ERK and JNK in bone marrow derived macrophages (Fig. R5e). These results suggest that TNF- α may not be an upstream signal for NLRP3 inflammasome in the intestinal lamina propria mononuclear cells. Probably due to the distinct microenvironment in the gut, lamina propria mononuclear cells cannot respond to TNF- α stimulation, which is different from that in the skin.

Figure R5. TNF- α does not act upstream of NLRP3 inflammasome in the intestinal lamina propria mononuclear cells. (a-d), the intestinal lamina propria mononuclear cells were isolated from both WT and *Nlrp3*^{R258W} mice and stimulated with heat-inactivated fecal bacteria (BAC, MOI=1: 20) or TNF- α (200ng/ml) for indicated time with or without ATP (5mM) pulse, the secreted IL-1 β was determined by ELISA (a), and the expression of Il6, Il1b and *Nlrp3* was assayed by Q-PCR (b-d). (e), bone marrow derived macrophages of WT mice were stimulated with 100ng/ml TNF- α for 10min, the phosphorylation of ERK and JNK was analyzed by western blot.

- While the authors provided convincing evidence that the *Nlrp3* R258W mutation gives a growth advantage to both *Clostridia* XIVa and *Lactobacillus murinus* that were found to promote Tregs differentiation, the use of electronic microscopy and of fluorescent in situ hybridization with specific probes targeting either *Clostridia* or *Lactobacilli* species would

help to determine at which age mutant mice showed a greater colonization with the aforementioned bacteria and where those are colonizing the intestinal mucosa for promoting Tregs induction.

Reply: Thanks for the suggestions. These suggested assays are good examples to thoroughly characterize certain bacteria species to validate their properties such as Treg induction in the intestine, but our current study does not intend to elaborate on any specific bacteria species. Rather, we want to emphasize a functional and topological structural shift of gut microbiota in the *Nlrp3*^{R258W} mice (Fig. 3), which conferred the ability of Treg induction (Fig. 7). We dissected a set of core 50 OTUs representing the most significantly changed phylotype bacteria between the WT and *Nlrp3*^{R258W} mice derived microbiota, in which we found coincidentally that 2 OTUs (OTU64 and OTU575) and 1 OTU (OTU9) enriched in mutant microbiota were previously reported as Treg-inducing bacteria, eg. Clostridia XIVa and Lactobacillus murinus, respectively (Atarashi et al 2011, Science, doi: 10.1126/science.1198469; Tang et al 2015, Cell Host & Microbe, doi: 10.1016/j.chom.2015.07.003). The spatiotemporal colonization of both bacteria in the WT mouse intestine had been studied intensively with the mentioned methods in previous studies (Atarashi et al 2011, Science, doi: 10.1126/science.1198469; Atarashi et al 2013, Nature, doi: 10.1038/nature12331).

Nevertheless, we do agree that characterization of these representative Treg-inducing bacteria colonization in the mutant mouse intestine would help to validate the NLRP3^{R258W}'s ability to shape microbiota. As suggested, to monitor the dynamic colonization difference of Clostridia XIVa and Lactobacillus murinus between the *Nlrp3*^{R258W} mutant and WT mice, we collected the feces of these mice at different time points from just weaning to adulthood. Then we directly compared the 16S rRNA V3-V4 reads of Clostridia XIVa and Lactobacillus murinus OTUs between these two mouse lines. As shown in Fig. R6a-6c on next page, the lactobacillus OTU9 kept staying at higher level in mutant mice from weaning to adulthood, while Clostridia XIVa OTU64 and OTU575 were gradually getting higher after weaning in mutant mice. Such patterns of colonization were in accordance with previous report by Atarashi et al showing that Lactobacillus murinus colonizes the mice early before weaning, while clostridia XIVa colonizes after weaning (Fig. S5 from Atarashi et al 2011, Science, doi: 10.1126/science.1198469). The Akkermansia OTU1 in Fig. R6d was used as control. These data indicated that NLRP3^{R258W}'s ability to reshape microbiota may not change the spatiotemporal colonization pattern of the affected bacteria, but may change their abundance and interactive network.

Figure R6. Dynamic changes of selected OTUs in WT and *Nlrp3*^{R258W} mice derived faeces collected at indicated time points.

REVIEWERS' COMMENTS:

Reviewer #1 (Remarks to the Author):

The authors have conducted significant new experiments, and clarified microbiota data results and overall analytic approaches including approaches to correct for multiple comparisons, in response to the prior review.

They now show new data that crossing the Nlrp3R258W mice to the Rag^{-/-} background leads to spontaneous colitis. This supports the authors' conclusion that cells in the mature T & B lymphocyte compartment, likely Tregs, constrain the development of colitis in the face of Nlrp3 activation.

I have no further comments or concerns.

Reviewer #2 (Remarks to the Author):

The authors provide satisfactory responses to the raised concerns and have improved their manuscript accordingly. The conclusions are now better supported by the presented data and have been toned down in some instances (e.g. "no pathologies").

Reviewer #3 (Remarks to the Author):

This revised version of the manuscript is greatly improved. The authors have done a nice job when addressing the major concerns brought up in my original review by adding supporting additional data.

Reviewers' comments:

Reviewer #1 (Remarks to the Author):

Canonical inflammasome activation via IL-18 and IL1B has been implicated in driving colitis in humans. IL-1R blockade has shown therapeutic benefit in case series. Conversely, a gain-of-function mutation in NLRP3 leads to inflammasome activation and systemic, but not gut, autoinflammatory disease. Prior studies utilizing Nlrp3 deficient mice have yielded conflicting results regarding a protective or detrimental role in colitis. This is highly significant, as both specific inflammasome inhibitors and downstream IL-1R blockade are under study for broader treatment of IBD.

The authors have addressed this by examining recovery from DSS colonic injury in mice carrying a gain-of-function mutation in Nlrp3. They show that these mice exhibit an increase in basal IL1b, but not IL-18, production, and a profound shift in basal fecal microbial composition, function, and interconnections. This is associated with protection from colonic injury following acute DSS administration. Several complementary approaches using co-housing, antibiotic administration, and colonization of germ-free mice are used to show that the microbiota of mice carrying mutated Nlrp3 exerts a protective effect, and compound transgenic mice are used to show that this effect upon the microbiota depends upon intact IL-1 signaling. Studies using RNASeq and PCR demonstrate an increased in anti-microbial peptide expression, suggesting a role in modulating the microbiota. Finally, studies utilizing CD25 blockade and quantification of colonic fox3+ Tregs are used to conclude that

microbiota-induced Tregs contribute to the amelioration of the colonic injury.

Prior studies using Nlrp3 deficient mice have implicated alterations in the enteric microbiota, IL-18 enhancement of epithelial wound healing, and IL-10 production by foxp3+ Tregs as potential mechanisms by which Nlrp3 might exert a beneficial effect in the setting of acute colitis. Conversely, with more chronic colitic injury a pro-inflammatory effect of Nlrp3 has been shown, potentially via IL-18 and IL-1b dependent epithelial injury and expansion of IFNG producing effector Th1 lymphocytes.

The current report is the first to my knowledge to test the effect of a gain-of-function mutation in Nlrp3 upon the response to acute colonic injury. It is therefore highly relevant to understanding the surprising lack of gut injury in patients with similar mutations. The main novel aspect of the study is in defining in depth the effects of the Nlrp3:IL1 axis upon the commensal flora, and demonstrating through several complementary approaches its protective properties in the setting of acute colonic injury. These results will be of great interest to the community, given the divergent results of prior studies using Nlrp3 deficient mice, and the fundamental importance of inflammasome biology for mucosal homeostasis and potential novel IBD therapies.

However, there are some concerns that limit support for the authors' conclusions.

- 1) The data regarding the basal effect of the Nlrp3:IL1 axis upon a protective microbiota

are convincing and novel. However, I am not sure that protective effects of Nlrp3:IL-18 following DSS injury have been excluded. It would have been presumed based on prior studies that this was the primary protective effect. It would be quite useful to formally exclude a role for IL-18, at least via blockade in the Nlrp3^{R258W} mice in the setting of DSS administration.

Reply: Thanks for the comment and suggestion. This is an important issue, as the role of Nlrp3:IL-18 axis in experimental DSS colitis and colon cancer had been reported to be protective (Zaki et al, 2010, *Immunity*, doi: 10.1016/j.immuni.2010.03.003; Zaki et al, 2010, *J Immunol*, doi: 10.4049/jimmunol.1002046). As a matter of fact, we have considered this point and crossed the *Nlrp3*^{R258W} mice to *Il18*^{-/-} background followed with DSS colitis experiment. We found that in the absence of IL-18, the *Nlrp3*^{R258W} mutation did not show any protective effect (Supplementary Figs. 6e-6f in our manuscript). So, there is no doubt that IL-18 is protective against colitis development in mice. Interestingly, however, in addition to a basal effect of the Nlrp3:IL-1 β axis upon a protective microbiota under steady state, our further results showed that the production of IL-18 from the gut of experimental mice was dependent on caspase-1, BUT NOT NLRP3 (Fig. 6d, text lines 315-332). Upon DSS treatment for 4 days, IL-18 production was significantly elicited in the gut epithelium, which was diminished in the absence of caspase-1 BUT NOT NLRP3; while in the lamina propria mononuclear cells, IL-18 remained at basal level (Fig. 6d, lower panel). These results suggest that during DSS injury, IL-18 from the epithelium responded quickly, which potentially mediated its protective effect. But the IL-18 production from epithelial cells must have been mediated by some inflammasomes expressed in the epithelium (such as NLRC4/NLRP6/NLRP9b), BUT NOT NLRP3, because NLRP3 is expressed in the lamina propria mononuclear cells, not in gut epithelial cells (Figs. 1a, 6a-6b, supplementary Figs 6a). Of note, during the revision of our current manuscript, using fractionated intestinal cells, a study showed very similar results with our data that NLRP3 is expressed selectively in the lamina propria, while IL-18 is expressed only in the epithelium both in mice and humans (Fig. 2a, Extended Figs. 3a, 3c and 10c from Zhu et al, 2017, *Nature*, doi: 10.1038/nature22967). The reason why *Nlrp3*^{R258W} x *Il18*^{-/-} mice lost resistance to DSS challenge was probably due to a more predominant and frontier role for IL-18 in epithelium than the NLRP3-R258W protein in the lamina propria, because in principle DSS colitis model is to injure the epithelium first, then allow commensal bacteria to activate immune cells in the lamina propria. In summary, our data don't exclude IL-18's essential role in protecting host against DSS colitis, but suggest that the effect of IL-18 is not a direct consequence of NLRP3 inflammasome activation in the intestine.

2) The RNASeq experiment could have been quite helpful in discovering the key host:microbe protective mechanisms, but ideally should be performed with a larger number of WT and Nlrp3R258W mice in order to permit a more comprehensive analysis of the colonic global pattern of gene expression to provide better insight into potential protective pathways. Simply show a few antimicrobial peptides is not convincing in terms of establishing a likely Nlrp3:IL1 mechanism for shaping the commensal flora. A more comprehensive gene expression analysis would then also permit a formal multivariate analysis of covariation of the microbiota and host genes, likely leading to more compelling protective pathways. For example, it appears quite likely that microbiota derived metabolites are exerting a beneficial effect on the wound healing response to the DSS injury, and studies to directly test that would have been of great interest.

Reply: Thanks for the comments. Our current study is focused on addressing the unique novel working model for the NLRP3 inflammasome in the intestine (e.g. selective regulation of IL-1 β but not IL-18); and on revealing an unprecedented microbiota-host interaction relationship, wherein the flexible microbiota actively regulates host immunity (enhancing Tregs) to compensate otherwise detrimental inflammation caused by host genetic defect (NLRP3^{R258W}). We appreciate very much of the suggestion that using RNA profiling in combination with microbiota analysis to dissect the correlated pathways as potentially more compelling mechanisms downstream of NLRP3: IL-1 β for reshaping microbiota. Moreover, the speculation that certain microbiota derived metabolites are mediators of inflammation resistance is quite reasonable. In fact, our preliminary result from a direct analysis of fecal and urine metabolites by NMR have shown notable difference between WT and mutant mice derived samples, and several promising metabolites have been revealed for further functional validation (see Fig. R1 on next page please). However, to reach a conclusive theory, both the aforementioned assumptions require a large body of *in vivo* and *in vitro* experiments, which is impossible to accomplish in the revision stage of our present work. So in our humble opinion, also considering the focus of the current study, the data showing that the antimicrobial peptides are the effectors downstream of NLRP3:IL-1 β to reshape microbiota are substantial. We think it is more practical to include further in-depth mechanistic study based on our preliminary data to address the relationship between metabolites and host gene regulations in a separate study in the future.

Figure R1. The metabolomic analysis of feces and urine from WT and *Nlrp3*^{R258W} mice. Feces (a) and urine (b) from WT and *Nlrp3*^{R258W} mice were analyzed with NMR, left panels show the OPLS-DA plot with R2X and Q2 value, P value was from CV-ANOVA test; right panels show the peaks of different metabolites between the two lines of mice, red peaks indicate metabolites that are significantly different (Two tailed t-test, $P < 0.05$).

3) The final data set regarding a potential protective role of microbiota induced foxp3+Tregs is not convincing. This would need to at least be repeated with a greater sample size to be sure that it is reproducible, with appropriate controls for the effect of the CD25 blockade. It would be surprising if the predominant protective mechanism in the setting of acute DSS injury involved Tregs. Alternately, this potential mechanism could be left for another study which included a more clearly effector T cell dependent model, such

as T cell transfer colitis.

Reply: The protective role of Tregs in the acute DSS colitis model had been reported previously (Atarashi et al 2011, *Science*, doi: 10.1126/science.1198469; Wang et al, 2015, *Mucosal Immunology*, doi:10.1038/mi.2015.20). Moreover, adoptive transfer of CD4⁺CD25⁺ Tregs had been shown to inhibit innate inflammatory responses in mice on *Rag* deficient genetic background (Maloy et al, 2003, *J Exp Med*, doi: 10.1084/jem.20021345). Nonetheless, we do agree that the acute DSS colitis model alone may not be enough to reflect the microbiota induced Tregs' anti-inflammatory effect. So we also considered the T cell transfer colitis model as suggested. Actually we had started the crossing of the *Nlrp3*^{R258W} mice to *Rag1*^{-/-} background about 2 years ago. Quite unexpectedly, but also reasonably, after crossing to *Rag1*^{-/-} background, the *Nlrp3*^{R258W} mice developed notable spontaneous colitis (Fig. 8, text lines 395-415). This is a strong evidence to support: (1) Tregs are indeed critical for the restriction of gut inflammation in the *Nlrp3*^{R258W} mice; (2) the *Nlrp3*^{R258W} mutation in the lamina propria of intestine is intrinsically pro-inflammatory, it is thanks to the local microbiota and Tregs that the alternative homeostasis gets established. In addition, anti-CD25 administration in our experiment was efficient to deplete Tregs in the lamina propria, which can be seen in Supplementary Fig. 7a.

4) The protective versus detrimental effect of Nlrp3 in the setting of DSS colitis has been remarkably sensitive to experimental protocols, but overall seems to favor a protective role with more acute injury as in the current study, and a detrimental role with more chronic injury. In that sense, it would be quite helpful to expose the Nlrp3R258W mice to a more chronic DSS protocol (as reviewed in Pellegrini et al, Canonical and Non-Canonical Activation of NLRP3 Inflammasome at the Crossroad between Immune Tolerance and Intestinal Inflammation", Table 1) to determine whether this leads to a shift to exacerbation of inflammation, as reported for Nlrp3 deficient mice, or whether the microbiota continues to exert a protective effect even under conditions of more chronic injury.

Reply: Thanks for the suggestion. Actually, the AOM+DSS induced colorectal cancer model, with lower amount (2%) and repeated 3 cycles of DSS administration, can be taken as a

good reflection of chronic colitis model (Allen et al, 2010, J Exp Med, doi: 10.1084/jem.20100050). In our experiment, the colitis severity developed in the first two cycles was significantly higher in the WT mice, as reflected by the weight loss curve and the early deaths of WT mice in the second cycle of DSS treatment (Fig. 2k-l). These results indicated a similar protective role of *Nlrp3*^{R258W} mutation in the chronic inflammation.

5) The microbial networks are quite interesting, but should please be provided in a larger font which is easier to read.

Reply: We have reformatted the figure as suggested.

6) The statistical analyses need to be clarified, as it appears that t-tests have been used when repeated-measures ANOVA, or simple ANOVA, would have been more appropriate for comparisons involving multiple groups. It does not appear that a correction for multiple comparisons has been applied when appropriate, and it would be surprising if most of the data were normally distributed as suggested.

Reply: Thanks for pointing out the mistake. After careful re-examination, we have done the statistical analysis more properly, details can be found in the relevant text and figure legend.

7. It would be helpful to show IL18 in Figure 1.

Reply: We have added IL-18 secretion in Fig 1b as suggested.

8. Why were female mice not studied?

Reply: The female mice in our hands tend to have mild disease in the DSS colitis model, thus

from the very beginning we kept using male mice to make the data consistent. Because gender difference also have potential to interfere the disease phenotype, as well as the microbiota constitution, it's more reasonable to exclude female mice.

9. A supplemental Table for the Fig. 3c OTUs would be helpful.

Reply: Supplemental table 2 has been added as suggested.

10. Please clarify the mouse diet used.

Reply: The diet clarification has been added in the methods section as suggested.

11. Figure 4f typo: blue lines "negative" correlations

Reply: This has been corrected, thanks.

Reviewer #2 (Remarks to the Author):

General comments:

The authors studies gut physiology in a mouse model bearing a point mutation with gain of function in NLRP3 which is a critical component of the NALP3 inflammasome. In the mouse they found signs of skin inflammation as in cryopyrin-associated periodic syndromes but no signs of colitis. In contrast, the authors describe a "stronger resistance to colitis and CRC" due to increased expression of antibacterial peptides and changes in the microbiota. The manuscript is interesting (despite the lack of a phenotype) and the

experiments are well performed (cage controls and littermate controls). Some data need to be added to make the manuscript completely convincing.

Specific comments:

1) The authors describe that in their model they only found NLRP3 expression in lamina propria cells. The show data on isolated cells. Some additional data would be helpful to support their finding. There are good reports indicating an important role of the NLRP3 inflammasome in intestinal epithelial cells (e.g. George X. Song-Zhao, et al., Kevin J. Maloy, *Irp3 Activation in the Intestinal Epithelium Protects against a Mucosal Pathogen; Mucosal Immunol.* 2014 Jul; 7(4): 763–774.) In fact, this group already has shown that NLRP3 activation protects from colitis. Some morphological data on the presence of NLRP3 expression would be helpful. Is it possible that the point mutation changes the expression pattern? Why would expression in IEC be absent? What was the change of IL-18 expression under inflammatory conditions?

Reply: These are all important questions. Indeed, there are a few studies showed that NLRP3 acted in the nonhematopoietic compartment to protect mice against colitis (Zaki et al, 2010, *Immunity*, doi: 10.1016/j.immuni.2010.03.003; Song-Zhao et al, 2014, *Mucosal Immunology*, doi: 10.1038/mi.2013.94). However, final conclusions from both of these studies were based on bone marrow transplantation (BMT) strategy only, which compromised the reliability of their conclusions. Actually, there is a separate study also using BMT in DSS colitis model, but claimed a major role for the NLRP3 inflammasome in the hematopoietic compartment (Allen et al, 2010, *J Exp Med*, doi: 10.1084/jem.20100050), which was totally opposite to the conclusion from the first two studies. Moreover, the first two studies didn't show any expression evidence for NLRP3 protein in the epithelium directly, despite that a clear ASC expression in the epithelium was shown (Song-Zhao et al, 2014, *Mucosal Immunology*, doi: 10.1038/mi.2013.94). Intriguingly, although they proposed a major role for NLRP3 in the epithelium, Song-Zhao et al found that the epithelium inflammasome activation required ASC but not NLRP3 (Fig. 5g in Song-Zhao et al, 2014, *Mucosal Immunology*, doi: 10.1038/mi.2013.94). Thus, the contradictory results in these

aforementioned studies and the intriguing observation that NLRP3 protein was dispensable for gut epithelium inflammasome activation highlight the current ill-characterization of where and how NLRP3 expresses and functions in the intestine.

Our data using the *Nlrp3*^{eGFP} reporter mice showed clear expression of NLRP3 in the CD11b⁺/CD11c⁺ lamina propria mononuclear phagocytes but not in the epithelium (Fig. 1a, Supplementary Fig. 1a). Our western blot also successfully detected NLRP3 protein in the lamina propria mononuclear cells from both *Nlrp3*^{R258W} and WT mice (Fig. 1c). Moreover, the Q-PCR analysis confirmed such expression pattern of NLRP3, and the *Nlrp3*^{R258W} mutation did not change the expression pattern (Figs. 6a-6b, Supplementary Fig. 6a). As for the suggestion that to provide some morphological evidence of NLRP3 expression in the gut, we tried to detect the eGFP fluorescence signal in situ from the colon cryosections of *Nlrp3*^{eGFP} reporter mice, and indeed found some dot-like signals in the junction-area between muscularis mucosae and lamina propria compartments (arrow heads in Fig. R2, see below), while the epithelium showed none of such signal. As a positive control, we found very bright and ubiquitous signal from a CMV-eGFP reporter mice colon. The results were showed in Fig. R2. Since our FACS, western and QPCR data are consistent and convincing, and we think the in situ signal of the cryosection is too weak to reach publication quality (probably due to the small number of LPMCs under homeostasis), so we decided to show the data only in this response letter.

Figure R2. In situ examination of NLRP3 expression in the mouse colon. CMV-eGFP (positive control), WT (negative control) and *Nlrp3*^{eGFP} mice colons were dissected and cryosectioned, the eGFP fluorescence was monitored, and the nucleus was visualized by DAPI with confocal microscopy. Scale bars, 50um. Arrow heads are indicating the positive signal for NLRP3 in the lamina propria.

The reason why NLRP3's expression is absent in the IEC is currently unknown. But during the revision of the current manuscript, using fractionated intestinal cells, another paper showed very similar results with our data that both mouse and human NLRP3 are expressed selectively in the lamina propria, not in epithelium; while IL-18 is expressed only in the gut epithelium (Figs. 1a, 6a-6b, Supplementary Fig. 6a in our manuscript; and Fig. 2a, Extended Figs. 3a, 3c and 10c from Zhu et al, 2017, Nature, doi: 10.1038/nature22967). Certainly such expression pattern of NLRP3 in the intestine is interesting, while the mechanisms behind require future study, which can be an independent project: expression regulation of NLRP3 in different populations of cells in the gut and related mechanisms.

The change of IL-18 and IL-1 β under inflammatory conditions also intrigued us, so we tested their secretion in mice after 4-day DSS treatment. We found that the cell compartment pattern in which they express didn't change under such inflammatory conditions, however, we did find an increase of both IL-18 in epithelium and IL-1 β in the lamina propria triggered by the DSS challenge, and only the secretion of IL-1 β depended on the NLRP3 inflammasome (Figs. 6c-6d, text lines 315-332).

2) As IL-18 has been shown to play an important role in mucosal defense mechanisms, the addition of data on IL-18 secretion would be important. As intestinal epithelial cells (IEC) are an important source of IL-18 and IL-18 is also processed by the NLRP3 inflammasome those data would also add to the question mentioned in point 1.

Reply: Thanks for the suggestion. Accordingly, in Fig. 1b, we have added the secretion of IL-18, showing its lack of regulation by *Nlrp3*^{R258W} mutation in the lamina propria. Our data in Figs. 6a-6d and in Supplementary Fig. 6b all supported the concept that IEC is the major source for IL-18, and IL-18 is not regulated by NLRP3, because NLRP3 is not expressed in IEC. These results are consistent with previous data, for example, Song-Zhao et al showed that both IL-18 secretion and inflammasome activation in IEC were independent from NLRP3 (Fig. 4a and Fig. 5g from Song-Zhao et al, 2014, Mucosal Immunology, doi: 10.1038/mi.2013.94). In addition, very similar with our data, Zhu et al showed that both mouse and human NLRP3 are expressed selectively in the lamina propria, while IL-18 is expressed only in the epithelium in the gut (Fig. 2a, Extended Figs. 3a, 3c and 10c from Zhu et al, 2017, Nature, doi: 10.1038/nature22967).

3) NLRP3 inflammasome and autophagy functions are closely connected. How was autophagy changes in the cells with over-activation of NLRP3?

Reply: This is an interesting question. As shown in the following Fig. R3, we found a dampened autophagy response in the peritoneal macrophages from our *Nlrp3*^{R258W} mice upon Rapamycin or LPS stimulation, which is reflected by the decreased LC3-II in the *Nlrp3*^{R258W} mutant cells than that in WT cells.

Figure R3. Autophagy induction in WT and *Nlrp3*^{R258W} peritoneal macrophages. The peritoneal macrophages were induced with i.p. injection of 2ml 3% thioglycolate broth per mouse for 4 days, the isolated cells were then stimulated with 1 μ M Rapamycin or 500ng/ml lipopolysaccharide (LPS) for 7hr in the absence or presence of 0.1 μ M Bafilomycin A, which was added 30min prior the stimulation. The cell lysates were collected for western blot detection of autophagy associated protein LC3-I/II. GAPDH was used as loading control.

4) It is impossible to say that the colon of the *Nlrp3*^{R258W} mice was “free of pathologies”.

With the methods used the authors did not find any alterations. However, there could be differences in innate immune functions that don't lead to any morphological changes. The statement needs to be toned down. What about IL-18 levels in the tissue?

Reply: Thanks for the comment. Accordingly, we have changed the term “free of pathologies” to a more moderate one “maintained homeostasis”. The tissue level of IL-18 under resting conditions has also been added in Supplementary Fig. 2g.

5) The authors only performed acute DSS colitis. A model for chronic intestinal inflammation would be chronic DSS colitis or spontaneous in *Nlrp3*^{R258W}/*Il10* deficient mice.

Reply: Thanks for the comment. Actually we have included some chronic intestinal inflammation model in our study: First, in the AOM+DSS induced colorectal cancer model, we treated mice with lower amount (2%) and repeated (3 cycles) administration of DSS, which can be taken as a chronic colitis model (Allen et al, 2010, J Exp Med, doi: 10.1084/jem.20100050). In this model, the colitis severity developed in the first two cycles was significantly higher in the WT mice, as reflected by the weight loss curve and the early deaths of WT mice in the second cycle of DSS treatment (Figs. 2k-2l). These results indicated a similar protective role of *Nlrp3*^{R258W} mutation in the chronic inflammation. Secondly, we also considered the T cell transfer colitis model as a more chronic model to explore whether without Tregs the *Nlrp3*^{R258W} mice would develop stronger disease in this model. Actually, we had started the crossing of the *Nlrp3*^{R258W} mice to *Rag1*^{-/-} background about 2 years ago. Quite unexpectedly, but also reasonably, after crossing to *Rag1*^{-/-} background, the *Nlrp3*^{R258W} mice developed notable spontaneous colitis (Fig. 8, text lines 395-415). This is a strong evidence to support: (1) Tregs are indeed critical for the restriction of gut inflammation in the *Nlrp3*^{R258W} mice; (2) the *Nlrp3*^{R258W} mutation in the lamina propria of intestine is intrinsically pro-inflammatory, it is thanks to the local microbiota and Tregs that the alternative homeostasis gets established. In addition, anti-CD25 administration in our experiment was efficient to deplete Tregs in the lamina propria, which can be seen in Supplementary Fig. 7a.

Together, by using these two different chronic colitis models, we are able to support our conclusion that *Nlrp3*^{R258W} mice maintain homeostasis and resist exogenous inflammatory insults through the reshaped microbiota-induced Tregs besides the same conclusions obtained with acute DSS colitis model.

6) A language editing would certainly be helpful.

Reply: Thanks for the suggestion. We have edited our manuscript with MSC-Scientific Editing service.

Reviewer #3 (Remarks to the Author):

In humans, dysregulation of inflammasome signalling by gain-of-function mutations in the coding region of NLRP3 results in a group of diseases called cryopyrin-associated periodic syndromes (CAPS), which are characterized by auto-inflammatory conditions in skin, joints and eyes. Intriguingly, severity of skin disease was linked to an exacerbated response of mast cells to the skin microbiota in the mice carrying a CAPS-causing mutation, while no gut symptoms were reported. The investigation of the underlying mechanism leading to either disease protection in the gut or susceptibility in the skin in response to a disease-causing Nlrp3 mutation is of great important. Collectively, the methodology used in this study is adequate and the conclusions reached in this article are well supported by experimental data. The authors provided experimental evidence that the absence of intestinal inflammation in the mutant mice was a consequence of a greater abundance of regulatory T cells in response to several changes in the composition of its gut microbiota. Mechanistically, the authors demonstrated that exacerbated interleukin-1 signaling activation in mice carrying the Nlrp3 R258W mutation results in downstream activation of several antimicrobial peptides and in subsequent changes in the composition of the gut microbiota that protects mice against intestinal inflammation. Here, there are some concerns that, I think, need to be considered:

- The authors showed a greater ability of colonic CD11b+/CD11c+ phagocytes to secrete IL1b in response to heat-inactivated fecal bacteria, but it is not yet clear if the latter responsiveness of phagocytes is acquired in response to age-induced changes in the composition of the gut microbiota (eg. after weening). We may indeed expect that the phenotype is constitutive as mutant mice develop skin disease in response to some

microbes when being neonates. Foremost, it would strengthen the manuscript to determine whether intestinal mast cells may secrete greater amount of IL1b in the colon of mutant mice as it was previously shown in the skin (Immunity. 2012 Jul 27;37(1):85-95).

Reply: Thanks for the comment and suggestion. For the question about whether the NLRP3 inflammasome hyperactivity of R258W mutant colonic CD11b⁺/CD11c⁺ phagocytes was a consequence of age-induced changes in the gut microbiota, we used 3-week-old R258W mutant and littermate WT mice to isolate their colonic lamina propria mononuclear cells, and we found that the responsiveness of these young mice derived cells to bacteria stimulation was already robust (Fig. R4a-4b on next page). This result supports the stated expectation and previous observations that neonatal CAPS mice develop skin symptoms already.

To evaluate the role of intestinal mast cells, we examined the NLRP3 expression in these cells. However, as shown in the following Figure R4c, we failed to detect a positive NLRP3 expression signal in these intestinal mast cells, whereas the signal in the lineage⁺ lamina propria cells was clear. This result suggested a limited, if any, role of NLRP3 inflammasome in the intestinal mast cells under resting status to shape gut microbiota.

Figure R4. NLRP3^{R258W} Inflammasome activity in intestinal lamina propria mononuclear cells from young mice upon weaning and in intestinal mast cells. (a-b), the intestinal lamina propria mononuclear cells were isolated from just weaned mice (3 weeks old) and stimulated

with heat-inactivated fecal bacteria (BAC, MOI=1: 20) with or without ATP(5mM) pulse, the secreted IL-1 β and TNF- α were determined through ELISA. (c) isolated colonic lamina propria mononuclear cells from mast cell deficient *Kit*^{W-sh/W-sh}, WT and *Nlrp3*^{eGFP} mice were stained with markers for mast cells (gated as CD45.2⁺lin⁻CD117⁺Fc ϵ r1 α ⁺ cells). On gated mast cells, the NLRP3 reporter eGFP signal was monitored.

- While the authors provide experimental evidence that IL1 signaling is required for protection against colitis by shaping the composition of the gut microbiota in mutant mice, additional experiments are required to determine the role of upstream since neutralization of TNF- α abrogated IL-1 β production and skin disease in neonatal *Nlrp3* mutant mice (Immunity. 2012 Jul 27;37(1):85-95).

Reply: To test whether TNF- α acts upstream of NLRP3 inflammasome in the lamina propria mononuclear cells, we directly applied TNF- α to stimulate these cells followed with ATP pulse. However, as shown in the following Figure R5, we didn't observe any activation of the NLRP3 inflammasome as indicated by IL-1 β secretion (Fig. R5a on next page). Examination of IL-6, IL-1 β and NLRP3 expression found minimal upregulation of these genes by TNF- α either (Figs. R5b-5d on next page). We validated that the recombinant TNF- α we used was functional, because it successfully stimulated the phosphorylation/activation of ERK and JNK in bone marrow derived macrophages (Fig. R5e). These results suggest that TNF- α may not be an upstream signal for NLRP3 inflammasome in the intestinal lamina propria mononuclear cells. Probably due to the distinct microenvironment in the gut, lamina propria mononuclear cells cannot respond to TNF- α stimulation, which is different from that in the skin.

Figure R5. TNF- α does not act upstream of NLRP3 inflammasome in the intestinal lamina propria mononuclear cells. (a-d), the intestinal lamina propria mononuclear cells were isolated from both WT and *Nlrp3*^{R258W} mice and stimulated with heat-inactivated fecal bacteria (BAC, MOI=1: 20) or TNF- α (200ng/ml) for indicated time with or without ATP (5mM) pulse, the secreted IL-1 β was determined by ELISA (a), and the expression of Il6, Il1b and *Nlrp3* was assayed by Q-PCR (b-d). (e), bone marrow derived macrophages of WT mice were stimulated with 100ng/ml TNF- α for 10min, the phosphorylation of ERK and JNK was analyzed by western blot.

- While the authors provided convincing evidence that the *Nlrp3* R258W mutation gives a growth advantage to both *Clostridia* XIVa and *Lactobacillus murinus* that were found to promote Tregs differentiation, the use of electronic microscopy and of fluorescent in situ hybridization with specific probes targeting either *Clostridia* or *Lactobacilli* species would

help to determine at which age mutant mice showed a greater colonization with the aforementioned bacteria and where those are colonizing the intestinal mucosa for promoting Tregs induction.

Reply: Thanks for the suggestions. These suggested assays are good examples to thoroughly characterize certain bacteria species to validate their properties such as Treg induction in the intestine, but our current study does not intend to elaborate on any specific bacteria species. Rather, we want to emphasize a functional and topological structural shift of gut microbiota in the *Nlrp3*^{R258W} mice (Fig. 3), which conferred the ability of Treg induction (Fig. 7). We dissected a set of core 50 OTUs representing the most significantly changed phylotype bacteria between the WT and *Nlrp3*^{R258W} mice derived microbiota, in which we found coincidentally that 2 OTUs (OTU64 and OTU575) and 1 OTU (OTU9) enriched in mutant microbiota were previously reported as Treg-inducing bacteria, eg. Clostridia XIVa and Lactobacillus murinus, respectively (Atarashi et al 2011, Science, doi: 10.1126/science.1198469; Tang et al 2015, Cell Host & Microbe, doi: 10.1016/j.chom.2015.07.003). The spatiotemporal colonization of both bacteria in the WT mouse intestine had been studied intensively with the mentioned methods in previous studies (Atarashi et al 2011, Science, doi: 10.1126/science.1198469; Atarashi et al 2013, Nature, doi: 10.1038/nature12331).

Nevertheless, we do agree that characterization of these representative Treg-inducing bacteria colonization in the mutant mouse intestine would help to validate the NLRP3^{R258W}'s ability to shape microbiota. As suggested, to monitor the dynamic colonization difference of Clostridia XIVa and Lactobacillus murinus between the *Nlrp3*^{R258W} mutant and WT mice, we collected the feces of these mice at different time points from just weaning to adulthood. Then we directly compared the 16S rRNA V3-V4 reads of Clostridia XIVa and Lactobacillus murinus OTUs between these two mouse lines. As shown in Fig. R6a-6c on next page, the lactobacillus OTU9 kept staying at higher level in mutant mice from weaning to adulthood, while Clostridia XIVa OTU64 and OTU575 were gradually getting higher after weaning in mutant mice. Such patterns of colonization were in accordance with previous report by Atarashi et al showing that Lactobacillus murinus colonizes the mice early before weaning, while clostridia XIVa colonizes after weaning (Fig. S5 from Atarashi et al 2011, Science, doi: 10.1126/science.1198469). The Akkermansia OTU1 in Fig. R6d was used as control. These data indicated that NLRP3^{R258W}'s ability to reshape microbiota may not change the spatiotemporal colonization pattern of the affected bacteria, but may change their abundance and interactive network.

Figure R6. Dynamic changes of selected OTUs in WT and *Nlrp3*^{R258W} mice derived faeces collected at indicated time points.